# A Novel Framework for Analyzing Rainy Season Dynamics in semi-arid environments: A case study in the Peruvian Rio Santa Basin

Lorenz Hänchen[1], Emily Potter[2], Cornelia Klein[3], Pierluigi Calanca[4], Fabien Maussion[5], Wolfgang Gurgiser[6], and Georg Wohlfahrt[1]

[1]Universität Innsbruck, Institut für Ökologie, Innsbruck, Austria
[2]University of Sheffield, School of Geography and Planning, Sheffield, UK
[3]UK Centre for Ecology and Hydrology, Wallingford, UK
[4]Agroscope Reckenholz, Zurich, Switzerland
[5]University of Bristol, School of Geographical Sciences, UK
[6]Universität Innsbruck, Institut für Atmosphären- und Kryosphärenwissenschaften, Innsbruck, Austria

**Correspondence:** Lorenz Hänchen (lorenz.haenchen@uibk.ac.at)

**Abstract.** In semi-arid regions, the timing and duration of the rainy season determines plant water availability, which directly impacts food security. Rainy season metrics, which aim to define and, in some cases, predict the onset and end of seasonal rains can support agricultural planning, such as scheduling planting dates and managing water resources. However, these metrics based on precipitation time series do not always accurately reflect plant water availability, and the variety of available metrics can complicate the selection of the most suitable one. Furthermore, a metric's ability to capture observed vegetation variability can indicate its applicability over larger spatial or temporal scales. This study introduces a new bucket-type metric that incorporates a simplified water balance, accounts for both accumulation and storage and also takes inter-annual legacy effects into account. We evaluate its performance against seven commonly used rainy season metrics, both calibrated and uncalibrated, using 18 years of satellite-derived Normalized Difference Vegetation Index from the semi-arid Rio Santa basin in the Peruvian Andes. Our results demonstrate that calibrating metrics using vegetation data significantly enhances their ability to capture rainy season dynamics, with the bucket metric outperforming others in both accuracy and robustness. Furthermore, we examine the sensitivity of all metrics to variations in rainfall intensity and frequency under future climate scenarios, using a previously published high-resolution dataset specifically designed for the Rio Santa basin which provides historical (1981–2018) rainfall data and future projections (2019–2100) based on 30 statistically downscaled CMIP5 models for RCP 4.5 and 8.5 scenarios respectively. While most rainy season metrics exhibit expected correlations in response to climatic changes, some established metrics display physically inconsistent behavior due to methodological artifacts, highlighting their limitations in assessing hydroclimatic changes. In addition to the sensitivity analysis, we evaluate long-term trends in rainy season characteristics. Statistically downscaled CMIP5 ensemble projections for the future period suggest only a slight delay in the rainy season end, with no consistent trends in onset timing. Instead, inter-annual variability and ensemble spread remain the dominant influences. Our findings emphasize the need for careful calibration of metrics across diverse climate scenarios and different locations to ensure

their reliability for agricultural planning, policymaking, and climate adaptation strategies. By providing a novel framework for evaluating rainfall metrics, this study offers a scalable approach that can be readily applied to other semi-arid regions.

## 1   Introduction

In semi-arid regions, people's livelihoods are closely linked to seasonal water availability, relying strongly on the timing of the rainy season (Warner et al., 2012). Forecasting the local to regional onset and end of the rainy season is a crucial requirement in agriculture, tourism, water management and hydro-electricity generation while changes to the timing of the rainy season are frequently used as a measure of climate change (e.g. Zampieri et al., 2023). Previously, a variety of approaches for numerically determining the onset and end of rainy seasons from precipitation time series have been used in regions with distinct seasonalities of rainfall (e.g. Bombardi et al., 2019b; Fitzpatrick et al., 2015; Sedlmeier et al., 2023). Broadly, these metrics consist of threshold-based approaches which must be configured for each region (Sedlmeier et al., 2023), or time series inflection point approaches (hereafter objective metrics) which, in theory, are applicable to any region with a distinct hygric seasonality (Liebmann et al., 2007; Liebmann and Marengo, 2001). The latter have been previously used to create a global dataset of rainy season dynamics (Bombardi et al., 2019a). Furthermore, specialized methods have been designed for regions with bimodal rainy seasons (e.g. Dunning et al., 2016) in mind or for regions with high spatiotemporal variability of rainfall by employing data manipulation approaches such as Principal Components Analysis (Camberlin and Diop, 2003), two-phase linear regression (Cook and Buckley, 2009) or a flexible definition of the hydrological year to account for spatial and interannual variability in certain regions (Ferijal et al., 2022; Seregina et al., 2018, to name a few).

The resulting onsets and ends of rainy seasons can vary considerably depending on whether the methods were tailored to specific rain-gauge data, crop requirements or larger-scale characterization of temporal monsoon developments (Fitzpatrick et al., 2015; Sedlmeier et al., 2023). Often, the importance of determining rainy season characteristics for either agricultural planning, monitoring of ecosystems, assessments of temporal water availability in the light of a changing climate or water management topics in general is emphasized (e.g. Bombardi et al., 2019b; Fitzpatrick et al., 2015). However, observational and gridded precipitation time series are typically subject to significant uncertainties, such as spatial representativeness issues, measurement errors (such as undercatch in windy conditions), temporal inconsistencies, and biases in retrieval algorithms (e.g. Kidd et al., 2017; Pollock et al., 2018). These uncertainties can lead water users and managers to make improper assumptions or take misguided actions. These uncertainties are particularly problematic in regions where decisions about planting, irrigation scheduling, or reservoir management rely heavily on short- or mid-term rainfall predictions. To the best of our knowledge, strategies to validate the outputs of rainy season metrics against independent observations are currently lacking. This raises concerns about whether such metrics accurately capture conditions on the ground and highlights the need for validation frameworks that ensures their relevance and reliability for practical applications and allows to reliably deduce climatic changes. Furthermore, other aspects such as legacy effects beyond one hydrological year or the sensitivity of rainy season metrics to the

alteration of the hydrological cycle, which is to be expected under climate change, have so far not been assessed.

This raises the challenge of designing an independent validation approach. While variables directly linked to the hydrological cycle such as soil moisture measurements would represent the ideal choice, their availability at (near-)climatological timescales is limited. In semi-arid regions, vegetation dynamics provide a useful alternative, as they exhibit a strong correlation with the seasonal precipitation cycle, albeit with a characteristic time lag (Hänchen et al., 2022). Remotely sensed proxies for vegetation development offer high spatiotemporal resolution and have been successfully used to study vegetation development for more

than half a century (starting with Rouse Jr et al., 1973). We therefore argue that incorporating an independent metric validation scheme based on vegetation development provides three crucial advantages: Firstly, validated and calibrated rainy season metrics align directly with local vegetation responses to changes in water availability. Secondly, time series of precipitation or other water-related variables can be tested regarding their quality. Lastly, previously published metrics, often designed for specific data and locations, can be assessed for their applicability in different regions. Spectral vegetation indices, which serve

as proxies for land surface greenness, are a promising candidate for calibrating rainy season metrics in semi-arid regions due to their high spatio-temporal resolution and availability from satellite data.

In this study, we develop and demonstrate a novel approach to calibrate rainy season metrics using vegetation dynamics, focusing on the Upper Rio Santa basin (also: Callejón de Huaylas) in the tropical Peruvian Andes. This region is characterized by high seasonal variability of precipitation with the majority of annual precipitation occurring between September and April and

little annual variability in temperature (see Figure 1 for the geographic location and a climograph). The region encompasses a complex hydroclimate system governed by the topography, the numerous abundance of glaciers on the Cordillera Blanca (eastern slopes of the valley), the temporal evolution of the South American monsoon (Espinoza et al., 2020; Garreaud, 2009; Klein et al., 2023a) and its interaction with ENSO (e.g. Hänchen et al., 2022; Maussion et al., 2015). In this region, a thorough understanding of the dynamics of the rainy season is crucial for regional water resources and agriculture, as the seasonal rain

provides water for irrigation, energy production, and the maintenance of ecosystems (e.g. Dextre et al., 2022; Drenkhan et al., 2022). There has been much attention on the past, present and future alteration of water availability in response to changes in glacial melt (e.g. Bury et al., 2010; Drenkhan et al., 2015; Fyffe et al., 2021). Small-scale farmers however often have no or limited access to glacier-fed river runoff and perceive increasing challenges related to precipitation seasonality (Gurgiser et al., 2016) and/or water quality (Rangecroft et al., 2023). Recently, more efforts to understand and monitor several aspects of

precipitation changes in the Rio Santa basin have been undertaken, but it remains challenging to derive successful mitigation strategies (Hänchen et al., 2022; Klein et al., 2023b; Mateo et al., 2022; Potter et al., 2023). Future climate scenarios indicate an overall increase in annual precipitation (Potter et al., 2023).

Potential shifts in the timing of the rainy season in the region, despite their profound implications for both societal and eco-

logical systems, have only recently been assessed. Notably, De la Cruz et al. (2025) used an objective metric to derive rainfall sums and rainy season onset and end for a Peru-wide network of meteorological stations based on statistically downscaled CMIP6 projections to derive future changes. They found an increase in future annual precipitation and similar to other studies

show that past rainy season dynamics in the broader Andean region reveal high inter-annual variability in rainy season onset, with generally non-significant or weak longer-term trends (Garcia et al., 2007; Giráldez et al., 2020; Gurgiser et al., 2016; Sedlmeier et al., 2023). Across those studies, the end of the rainy season is notably less variable than the start, while showing no or small significant changes historically. For the Rio Santa basin specifically, Hänchen et al. (2022) note a delayed end of the growing season between 2000 and 2020, indicating increased water availability difficult to detect from both satellite or gauge rainfall data due to the small rainfall totals during the early dry season. Additionally, the regional hydroclimate experiences a complex interaction with El Niño Southern Oscillation (ENSO), where the overall amount of rainy season precipitation, in most, but not all years, increases (decreases) with La Niña (El Niño) (e.g. Maussion et al., 2015; Vuille et al., 2008). At the same time, there are indications that El Niño conditions might cause seasonal rainfalls to start earlier, thus increasing overall plant water availability even though peak season rainfalls are reduced (Hänchen et al., 2022). This response contrasts with other basins in proximity, such as the Mantaro River basin, where the opposite pattern has been suggested (Giráldez et al., 2020) thus highlighting the spatial heterogeneity of hydroclimatic responses within the Andes.

To account for these difficulties, we employ a multi-faceted approach capitalizing stem on previous studies: We combine several precipitation datasets with remote sensing data on temporal vegetation development. Specifically, we calculate the rainy season metrics based on convection-permitting, bias-corrected Weather Research and Forecasting (WRF) precipitation data and statistically downscaled CMIP5 projections (Potter et al., 2023) and use CHIRPS gridded data (Funk et al., 2015) as well as data from three local weather stations for comparison. For validation and calibration, we utilize Land Surface Phenology (LSP) data for the period 2000–2018 and the spatial extent of the Rio Santa basin, derived from the temporal development of the remotely sensed Normalized Difference Vegetation Index (NDVI) provided by Hänchen et al. (2022). Their research demonstrated that NDVI — an indicator of vegetation greenness available at high spatio-temporal resolution — captures variability and changes in water availability in the semi-arid Rio Santa basin, where water availability is the primary limiting factor for plant growth. This high spatial resolution is shown in Fig. 1, which shows the 2000–2018 average NDVI for the Rio Santa basin, illustrating both longitudinal and altitudinal gradients. Similarly, other studies have demonstrated the applicability of NDVI in understanding precipitation variability in the central Peruvian Andes (Quiroz et al., 2011; Yarleque et al., 2016).

The principal objective of this study is to showcase a novel framework for characterizing the rainy season, emphasizing the importance of employing a calibration strategy for inferred rainy season onsets and ends. In addition, we test the sensitivity of rainy season metrics to plausible changes in rainfall intensity and frequency, as might occur due to global warming. By capturing shifts in seasonal rainfall dynamics, our approach provides a foundation for identifying and understanding hydrological changes that may inform future adaptation strategies. The proposed framework is designed to improve our understanding of variations in water availability within semi-arid regions, offering insights that extend beyond the Rio Santa basin and can be applied to similar climates. Regarding the Rio Santa basin, we aim to provide insights into past and future changes. We achieve this by:

1. Establishing an approach to derive reliable rainy season metric outputs from several existing methodologies from precipitation time series by calibrating them using LSP data.

2. Introducing a novel methodology to the community to derive rainy season indicators, where we simulate water availability in a simplified fashion using only precipitation time series as input and a number of calibrated constants.

3. Testing the response and sensitivity of each metric to physically plausible changes in the rainy season.

4. Analyzing changes of the temporal evolution of the rainy season in the Rio Santa Basin. By making use of the aforementioned calibrated metrics, we explore past (since 1981) and future (until 2100) changes of the onset and end of the rainy season based on CMIP5 models for the region.

125

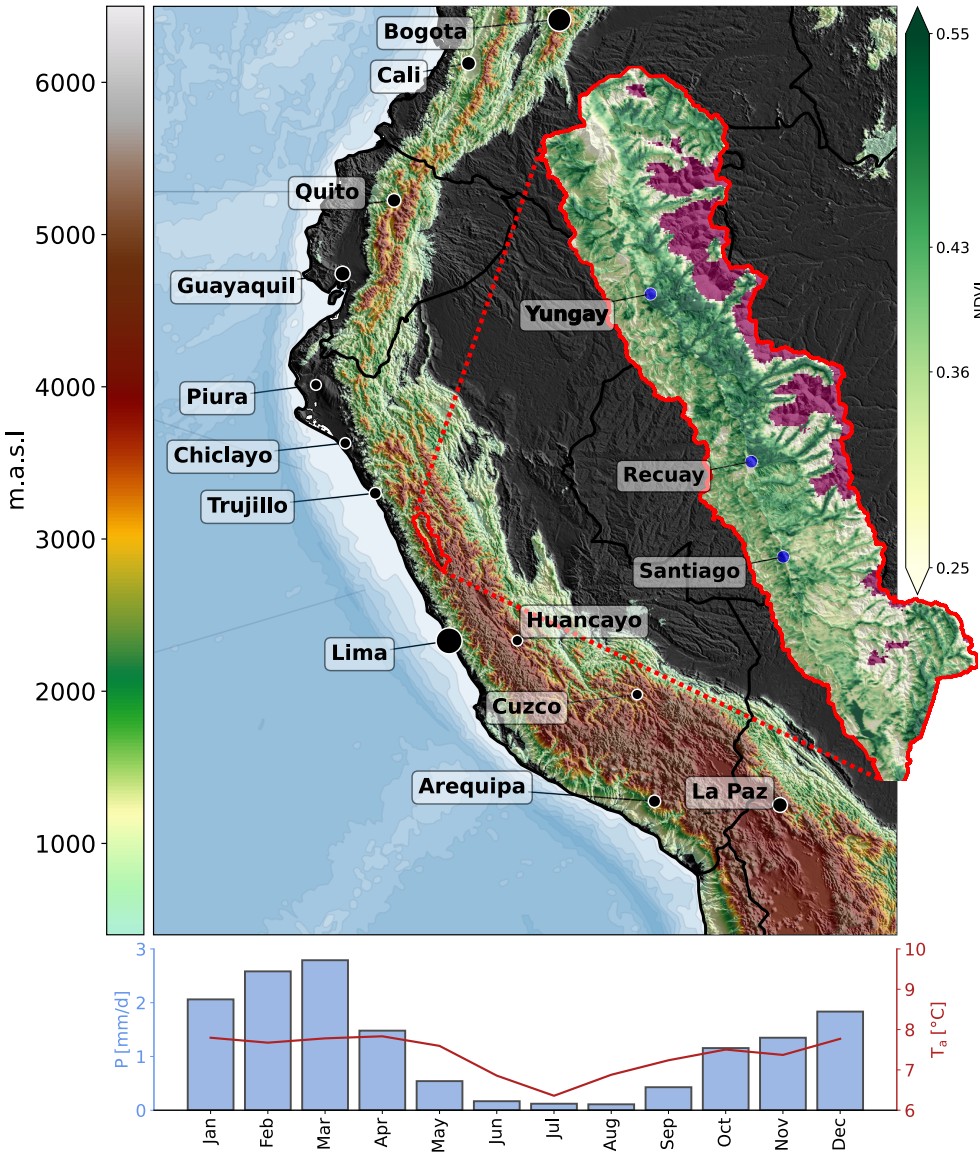

**Figure 1.** Overview of the study area. The large map shows the topography of the Andes with elevations below 500m shown in black (based on SRTM data, USGS EROS Archive, 2021), administrative borders and larger towns in the greater region. The enlarged area of the Rio Santa basin shows the long-term (2000-2018) average NDVI of each pixel; no-data areas, mostly referring to land-covers such as Ice or Bare rocks are shown in magenta color. The three blue dots indicate the locations of the local weather stations used in this study. Additionally, the Climograph at the bottom shows the seasonality of precipitation and temperature derived from spatially averaged WRF data for 1981-2018 (Potter et al., 2023) for the Rio Santa basin.

## 2 Material and methods

### 2.1 Data

As a target dataset for calibration, we use Land Surface Phenology (LSP) data by Hänchen et al. (2022) from 2000 to 2018 derived from MOD13Q1 and MYD13Q1 (Didan, 2015a, b) NDVI time series in 250 m spatial resolution (c.f. Fig. 1). The data were i) filtered on quality assurance criteria, ii) gap-filled and smoothed using a Gaussian process regression algorithm (Belda et al., 2020), iii) masked based on unimodal seasonal vegetation development and land-cover data to exclude pixels which are evidently decoupled from the rainy season. LSP was assessed by applying a relative threshold to the Rio Santa basin average seasonal cycle of vegetation greenness (c.f. Caparros-Santiago et al., 2021) to obtain the start and end of the growing season based on which we subsequently calibrate all rainy season metrics. Specifically, the start (hereafter $SOS_{NDVI}$) and end (hereafter $EOS_{NDVI}$) of the growing season were derived as the day where the processed NDVI data reaches 30% of its seasonal amplitude (Hänchen et al., 2022). The resulting LSP data were averaged to the extent of the Rio Santa basin.

In our analysis, we utilize multiple precipitation datasets. A key component is the WRF bias-corrected regional climate model data published by Potter et al. (2023), which provides consistent precipitation estimates at 4 km grid spacing from 1981 to 2018. As a second gridded precipitation dataset for the recent past, we use the gridded Climate Hazards InfraRed Precipitation with Station data (CHIRPS, Funk et al., 2015), which are provided in $0.05° \times 0.05°$ spatial and daily temporal resolution between 1981 and 2018. The gridded data are compared to an average of three local weather stations (AWS) operated by the National Meteorological and Hydrological Service of Peru (SENAHMI), which were sufficiently gap-free. These three stations located at Yungay (9.14 °S, 77.75 °W), Recuay (9.73 °S, 77.45 °W) and Santiago (9.52 °S, 77.52 °W) are all located along the valley floor of the Rio Santa basin (Hunziker et al., 2017). In addition, Potter et al. (2023) produced statistically downscaled projections based on a 30-member CMIP5 ensemble from 2019 to 2100 using quantile delta mapping for both the Representative Concentration Pathways (RCP) 4.5 and 8.5 scenarios. These scenarios represent different greenhouse gas concentration trajectories, where the number indicates the associated radiative forcing in 2100 (in $Wm^{-2}$). RCP4.5 is a stabilization scenario with moderate mitigation efforts, while RCP8.5 represents a high-emission, business-as-usual trajectory. These data preserve CMIP5 model trends while adjusting precipitation magnitude and the number of wet days based on the bias-corrected WRF data, and are available in the same 4 km grid spacing from 2019 to 2100. The two RCP scenarios allow us to assess multiple trajectories of future changes in the rainy season in the Rio Santa basin and provide a large dataset for metric sensitivity analysis. We do not evaluate metrics for raw, coarse-scale CMIP data in this study, as at their native resolution, they are known to inadequately represent orographic processes and interannual variability (e.g. Gutierrez et al., 2024).

Both gridded precipitation datasets were restricted to the geographical coverage of the available NDVI pixels as seen in Fig. 1 within the Rio Santa basin to acknowledge that high precipitation sums in the elevated Cordillera Blanca regions (e.g., glacierized or bare ground land-cover) do not align with vegetation responses and then spatially averaged (i.e. no spatial dimension). We excluded leap year days (29[th] February) and performed the analysis based on a hydrological year definition suitable

for the Rio Santa basin starting from the 1st Sep and ending on the 31st of Aug of the subsequent year.

As vegetation responses to rainfall are not necessarily immediate, the lag between the NDVI and the precipitation data must be accounted for to allow to use the LSP data as targets. We therefore determined this lag between the spatial averages of smoothed NDVI and a 12-week rolling average for each of the three precipitation time series by utilizing a cross-correlation function to identify the lag with best alignment by the index (days) of the highest Pearson correlation coefficient. Finally, we
subtracted the determined lags for each precipitation dataset from the LSP data before further analysis (c.f. Fig. A1).

## 2.2 Rainy season metric calculation

Here, we apply the same threshold-based metrics which Sedlmeier et al. (2023) compiled for the Southern Peruvian Andes, hereafter called Gurgiser (Gurgiser et al., 2016), Climandes (Sedlmeier et al., 2023), Garcia (Garcia et al., 2007), FP (Frere and Popov, 1986) and JD (Jolliffe and Sarria-Dodd, 1994) and tune them specifically for the Rio Santa basin and each precipitation
dataset. The rationale of each rainy season metric can be found in Table 1. The first four metrics (Gurgiser, Climandes, Garcia and JD) to derive the rainy season onset (hereafter RSO) all use some combination of four conditions: 1) The day of the onset has to have precipitation above a threshold value; 2) The total precipitation in a defined period after the onset must exceed a certain sum; 3) There must be a minimum number of wet days in a defined period after the onset; 4) There must be no continuous periods of dry days over a certain length within a defined period after the onset. Gurgiser uses conditions 1, 2 and
3, Climandes conditions 1, 2 and 4, Garcia conditions 2 and 4, and JD conditions 3, 2 and 4. FP use a different approach by dividing a 30-day period into terciles, where each tercile must exceed a certain total precipitation, similar to condition 2 of the other metrics. For calibration, our implementation of FP involves examining the first, second, and third tercile, while adjusting the length of the period and total precipitation thresholds.

While the FP and JD metrics are focused exclusively on the onset of the rainy season, the three remaining threshold-based metrics provide a more comprehensive approach by also addressing the end of the rainy season (hereafter RSE): 1) defining a precipitation threshold for the potential day of the rainy season end and 2) defining a threshold for the precipitation sum over a number of subsequent days. The Garcia metric omits the first criterion. For comparison, we also tested two other, non-threshold-based metrics: The widely established metric by Liebmann and Marengo (2001), hereafter named LM, which
accumulates seasonal rainfall against the average of the hydrological year. Then, the days of the minimum and maximum are considered the onset and end of the rainy season. The method by Cook and Buckley (2009), hereafter CB, employs a change-point detection method, fitting a two-phase linear regression iteratively over i) the first 250 and ii) the last 200 days of the hydrological year independently. By minimizing the sum of squares of residuals, the best fit for the regressions is found and the changepoints determine the onset and end of the rainy season. We implemented this approach using the python package
pwlf (Jekel and Venter, 2019). Approaches considering data other than rainfall time series, combining threshold-based approaches with fuzzy-logic (Laux et al., 2008) or Pentad-based approaches (e.g. Giráldez et al., 2020; Marengo et al., 2001) are

beyond the scope of this study and thus not included.

| Name (Reference) | RSO (original) | RSO (adapted) | RSE (original) | RSE (adapted) |
|---|---|---|---|---|
| **Gurgiser** (Gurgiser et al., 2016) | $P_d > 0\,mm$ <br> $\sum P_{d:d+7} > 10\,mm$ <br> $N[P_{d:d+30} > 0\,mm] \geq 11$ | $P_d > \alpha_p^1$ <br> $\sum P_{d:d+\alpha_d^3} > \alpha_p^2$ <br> $N[P_{d:d+\alpha_d^5} > \alpha_p^6] \geq \alpha_d^4$ | $P_d = 0\,mm$ <br> $\sum P_{d:d+46} < 10\,mm$ | $P_d \leq \alpha_p^1$ <br> $\sum P_{d:d+\alpha_d^3} < \alpha_p^2$ |
| **Climandes** (Sedlmeier et al., 2023) | $P_d > 1\,mm$ <br> $\sum P_{d:d+5} \geq 8\,mm$ <br> $N_c[P_{d:d+30} < 0.1\,mm] \leq 6$ | $P_d > \alpha_p^1$ <br> $\sum P_{d:d+\alpha_d^3} \geq \alpha_p^2$ <br> $N_c[P_{d:d+\alpha_d^5} < \alpha_p^6] \leq \alpha_d^4$ | $P_d \leq 1\,mm$ <br> $\sum P_{d:d+30} < 16\,mm$ | $P_d \leq \alpha_p^1$ <br> $\sum P_{d:d+\alpha_d^3} < \alpha_p^2$ |
| **Garcia** (Garcia et al., 2007) | $\sum P_{d:d+3} > 20\,mm$ <br> $N_c[P_{d:d+30} < 0.1\,mm] \leq 9$ | $\sum P_{d:d+\alpha_d^2} > \alpha_p^1$ <br> $N_c[P_{d:d+\alpha_d^4} < \alpha_p^5] \leq \alpha_d^3$ | $P_{d:d+20} \leq 2^*\,mm$ | $P_{d:d+\alpha_d^2} \leq \alpha_{pr}^1$ |
| **FP** (Frere and Popov, 1986) | $\sum P_{d:d+10} \geq 25\,mm$ <br> $\sum P_{d+10:d+20} \geq 20\,mm$ <br> $\sum P_{d+20:d+30} \geq 20\,mm$ | $\sum P_{d:d+\frac{1}{3}\alpha_d^2} \geq \alpha_p^1$ <br> $\sum P_{d+\frac{1}{3}\alpha_d^2:d+\frac{2}{3}\alpha_d^2} \geq \alpha_p^3$ <br> $\sum P_{d+\frac{2}{3}\alpha_d^2:d+\alpha_d^2} \geq \alpha_p^4$ | Onset only | Onset only |
| **JD** (Jolliffe and Sarria-Dodd (1994), originally based on Stern et al. (1981)) | $N[P_{d:d+5} > 0.1\,mm] \geq 3$ <br> $\sum P_{d:d+5} \geq 25\,mm$ <br> $N_c[P_{d:d+30} < 0.1\,mm] \leq 7$ | $N[P_{d:d+\alpha_d^2} > \alpha_p^1] \geq \alpha_d^3$ <br> $\sum Pr_{d:d+\alpha_d^2} > \alpha_p^4$ <br> $N_c[P_{d:d+\alpha_d^6} < \alpha_p^1] \leq \alpha_d^5$ | Onset only | Onset only |
| **LM** (Liebmann and Marengo, 2001) | $d$ such that $A_d = \min A$ where <br> $A = \sum_{n=1}^{d} P_n - P$ | | $d$ such that $A_d = \max A$ where <br> $A = \sum_{n=1}^{d} P_n - P$ | |
| **CB** (Cook and Buckley, 2009) | Linear regression fitting and change point detection to determine when daily precipitation changes from decreasing each day to increasing each day (onset) and from increasing to decreasing (end). | | | |

**Table 1.** Rainy season metric rationales where $d$ is the day of the year marking the onset or end of the rainy season, $P_d$ is the precipitation on day $d$ (in mm), and for example, $\sum P_{d:d+6}$ is the sum of precipitation on each day from the onset to 6 days after the onset. $N[P_{d:d+30} > 0mm]$ is the number of days with precipitation over 0 mm in a 30 day period. Some metrics use $N_c$ instead of $N$, which represents continuous dry days, e.g. $N_c[P_{d:d+30} < 0.1\,mm] < 7$ is the condition that no dry spells of more than 7 days occur in the 30 days after the onset. The parameters $\alpha^1$ to $\alpha^n$ are the tuneable parameters of each metric, with $\alpha_p$ denoting a precipitation threshold in mm and $\alpha_d$ denoting an integer number of days. $A$ is the cumulative sum of anomalous precipitation from day 1 to $d$ and $P$ is the annual average daily precipitation. $^*$ For $RSE_{Garcia}$, which is published as $P_{d:d+20} = 0\,mm$ we used a value of 2 mm instead as 20 consecutive days with zero precipitation are not present in any of the datasets.

## 2.3 A new rainy season metric: The "bucket" metric

Finally, we introduce a novel approach, which attempts to simulate a simplified water balance by consecutive balancing of daily input through rainfall and output through constant evapotranspiration, additionally constrained by a minimum and maximum bucket water content, ensuring realistic water balance limits:

$$
BWC(t) = \begin{cases} BWC(t-1) + \frac{BD}{\rho} \cdot (P(t) - ET), & \text{if } BWC_{\text{mn}} \leq BWC(t) \leq BWC_{\text{mx}} \\ BWC_{\text{mn}}, & \text{if } BWC(t) < BWC_{\text{mn}} \\ BWC_{\text{mx}}, & \text{if } BWC(t) > BWC_{\text{mx}} \end{cases} \tag{1}
$$

where BWC [m$^3$/m$^3$] represents the Bucket water content at time t, BD is the bucket depth [m] and $\rho$ is the water density, here constant as 1000 kg/m$^3$], P [mm/day] is the precipitation input and ET [mm/day] is the daily output.

Note that ET [mm/day] is inspired by evapotranspiration but does not represent the actual physical process as the simplistic design of the metric considers it to be constant over the whole hydrological year, partly integrates other hydrological components such as runoff and thus within- and between seasonal variation is not directly accounted for. The metric starts at day d = 0 at an initial BWC (BWC$_{\text{ini}}$). The model is constrained as no further evaporation occurs as soon as a minimum value (BWC$_{\text{mn}}$) is reached. Similarly, a maximum value (BWC$_{\text{mx}}$) is defined where no more water is accumulated – any surplus conceptually runs off or drains from the bucket. The parameters BWC$_{\text{in}}$, BWC$_{\text{mx}}$, BWC$_{\text{mn}}$, t$_{\text{RSO}}$, t$_{\text{RSE}}$, BD and ET need to be tuned and do not change over time. For each season, rainy season onset and end are then determined based on two previously calibrated BWC thresholds denoted as t$_{\text{RSO}}$, t$_{\text{RSE}}$ in Figs. 2 & 3. In Fig. A2, an example of a full BWC and precipitation time series, along with the derived RSOs and RSEs, is shown.

In contrast to the metrics previously introduced which calculate each season independently, the bucket metric is able to calculate over the complete multi-year time series, allowing to incorporate legacy information about water availability prior to the rainy season of interest. As for the other approaches, we optimized all parameters according to the corresponding input data. While this approach is inspired by existing simple hydrologic bucket models and thus by actual hydrological processes, our aim is not to accurately represent these, but rather to account for parameters altering plant available water in a simplified fashion.

## 2.4 Calibration of threshold-based metrics

Using the NDVI-derived SOS$_{\text{NDVI}}$ and EOS$_{\text{NDVI}}$ as targets, we first tested the initial parameters provided by the corresponding authors. We then calibrated each threshold-based metric, along with our novel metric, by adjusting their parameters (c.f. Table 1) for each of the three precipitation time series. This was done using a Differential Evolution optimization algorithm (Storn and Price, 1997), with parameters constrained to physically plausible boundaries. Due to the limited number of growing seasons available (i.e., 18), splitting the data into calibration and validation periods would not have allowed to obtain robust

correlation and was therefore omitted. To allow for the robust and efficient processing of a large number of timeseries regarding the threshold-based metrics, we generally start the iterative search for the RSE starting from the previously derived RSO. Additionally, when a RSE is found within the 90 subsequent days following the RSO, the iterative search is continued to account for erroneous detection of dry spells in the early rainy season. For all metrics, RSEs were discarded if an unrealistically early end (before February 1$^{\text{st}}$) occurred.

## 2.5 Sensitivity analysis

To understand how tuned threshold-based metrics and objective methods respond to hydro-climatological changes, we utilize the large number of rainy seasons ($\sim$ 5000) provided by the future projections of Potter et al. (2023) to assess sensitivities of the metrics in regard to potential and physically plausible changes of the rainy season. To account for a variety of scenarios, we correlate the rainy season metric outputs (RSO/RSE) calculated for all rainy seasons independent of model, year or scenario with both full hydrological year and sub-seasonal (SON, DJF, MAM, JJA) rainfall sums. The sub-seasonal rainfall sums are referring to the same rainy season RSO/RSE were derived on, meaning that, for example, JJA refers to the dry months after the RSO. The rationale for correlating the seasonal rainfall sums even beyond the period where the RSO typically occurs is to test whether some of the metrics show implausible sensitivities which reduce their usefulness from a practical perspective. Additionally, we used four Expert Team on Climate Change Detection and Indices (ETCCDI) climate indices (Zhang et al., 2011) based on the WRF and statistically downscaled CMIP5 data created by Potter et al. (2023). These are the number of dry and wet days (DD, WD), defined as days with precipitation less and greater than 1 mm; the Simple Precipitation Intensity Index (SDII), representing the average daily precipitation on wet days (WD) and the sum of precipitation above the 95$^{\text{th}}$ percentile relative to the historical (1980-2018) period (R95pTOT). Using LSP data as dependent and rainfall sums as independent variables we assess sensitivities by applying bin-weighted linear regression for each variable independently. Bin sizes for the regressions are determined in an objective fashion by applying the Freedman-Diaconis rule (Freedman and Diaconis, 1981) to each of the nine independent variables.

## 2.6 Future projections

To reliably determine trends in future CMIP5 projections, we first excluded individual models for each rainy season metric if they produced five or more invalid values out of 81 seasons (2019–2100). Invalid values occurred when the conditions for RSO or RSE were not met within a given hydrological year. For the remaining data, we calculated linear trends separately for the historical WRF and CHIRPS datasets, as well as for both RCP scenarios of the CMIP5 ensemble, using linear regression. Trend significance was assessed using a Wald Test, with the null hypothesis that the slope is zero.

# 3 Results & Discussion

## 3.1 Evaluation of Rainy Season Metrics

We first compare the skill of all considered metrics in predicting the RSO close to respective reference $SOS_{NDVI}$ across all years and datasets. Figure 2a-e illustrates that all calibrated threshold-based metrics consistently predict the lag-corrected $SOS_{NDVI}$ across the three precipitation datasets, demonstrating a strong correspondence and outperforming the initial (i.e. uncalibrated, $INIT_{WRF}$) setup of the metric, which for all threshold-based metrics except JD is lacking correlation and showcasing higher RMSE values. The bucket metric stands out by exhibiting low errors (average RMSE = 8.7 days) and a robust correlation ($r^2$ =

0.79, on average) across all three input datasets (c.f. Fig. 2, Fig. 4). This is likely related to the fact that the bucket metric was designed to directly determine the lag between rainfall inputs and vegetation responses while the other metrics make use of the cross-correlation maximization. Hence, the bucket metric does account for legacy information between seasons. Although we did not directly observe a deterioration of correlation when removing the legacy information from the metric before parameter optimization, the resulting BWC timeseries was highly unrealistic and unsuitable for transferability (not shown). LM and CB

on the other hand demonstrate a relatively low agreement with $SOS_{NDVI}$ with LM showing weak correlations which for CB are missing (c.f. Fig. 2g and h).

Regarding the RSEs, the three threshold-based metrics (Fig. 3a-c) demonstrate a relatively low RMSE (ranging between 8.8 and 14.4 days) albeit lacking correlation (maximum $r^2$ = 0.25), whereas the bucket metric (Fig. 3d) shows an even lower

RMSE (5.6 days on average) and a weak correlation ($r^2$ = 0.4 on average), likely related to the bucket metric incorporating non-plant-available water simulated as bucket overflow (see Section 2.3). Gurgiser and Climandes share the same criteria for RSE and thus the resulting calibration is identical (c.f. Table 1 & Fig. 3). The LM and CB metrics show an overall low skill in predicting the lag-corrected $EOS_{NDVI}$ with high errors and LM showing a weak correlation for 2 of 3 datasets (c.f. Fig. 3f). The overall discrepancy across the metrics between skill on predicting $SOS_{NDVI}$ and $EOS_{NDVI}$ (see Fig. 4) may be linked to

$EOS_{NDVI}$ displaying low variability (standard deviation, $\sigma$ = 6.63 days), unlike the higher variability of $SOS_{NDVI}$ ($\sigma$ = 17.61 days). Additionally, the coupling between precipitation and water availability tends to be more prominent at the onset of the rainy season due to depleted hydrological system storages, resulting in reduced predictive power of rainfall for vegetation development as rainfalls recede.

While each metric shows reasonably high skill for all three precipitation datasets after calibration, the substantial differences in the resulting optimization parameters (c.f. Tables in Fig. 2 a-f & 3 a-d) underscore the necessity of tuning and testing rainy season metrics according to local climatic conditions, specific datasets and target applications. Given proper tuning, the results are comparable even though the metrics follow different logic and use a different number of parameters. Interestingly, among the threshold-based metrics, those with more parameters do not necessarily perform better in terms of error and correlation.

For example, $RSO_{FP}$ and $RSE_{Garcia}$, which have the fewest parameters (four and two, respectively), still show a consistent performance. A systematic test of the relevance of individual parameters is beyond our scope here, especially given the high

performance of the bucket metric which is our primary focus here. Our generally skilful results after calibration also illustrate that existing concerns (e.g. MacLeod, 2018) regarding the sensitivity of thresholds-based metrics to rainfall dataset bias and resolution appear to be surmountable if independent reference data are taken into account, rendering these metrics more flexi-
ble than is currently appreciated.

Although other authors asserted a strong agreement between the LM method and local threshold-based approaches (e.g. Dunning et al., 2016), our results emphasize that agreement in metric outputs from the same time series alone does not necessarily guarantee the representativeness towards plant growth or suitability for practitioners of any kind. While we acknowledge
the effectiveness of the LM method in larger scale climatological rainfall analysis, our analysis shows that it a) exhibits less correspondence with vegetation development than calibrated methods (Figs. 2–4) and b) tends to produce delayed onsets in the specific setting of the Rio Santa basin during extended dry spells following early-season rains (not shown). Similarly, the two-phase regression method (CB) tends to compute late onsets in cases of prolonged dry spells and/or when the development of the rainy season follows a non-linear trajectory (i.e. rainfall increase from the onset towards the peak of the rainy season), making
it unsuitable for accurate onset and end determination in the Rio Santa basin in many seasons. Furthermore, the objectivity of this method is limited as the rainy season needs to be split into two sub-seasons, potentially affecting resulting values. Here, we followed the same approach as the original authors, using the first 250 and the last 200 days of each season to determine the dates. Similarly, the metric demonstrates sensitivity to the definition of the hydrological year. For instance, shifting the start of the hydrological year back by two months significantly enhances the correspondence between $RSO_{CB}$ and $SOS_{NDVI}$,
while concurrently diminishing it between $RSE_{CB}$ and $EOS_{NDVI}$ (see Fig. A3 for an example). Given the high variability of the rainy season onset in the tropical Andes, coupled with the aforementioned sensitivity to the climatological year definition, we believe it is advisable to employ a flexible hydrological year approach (c.f. Ferijal et al., 2022) when exploring this method.

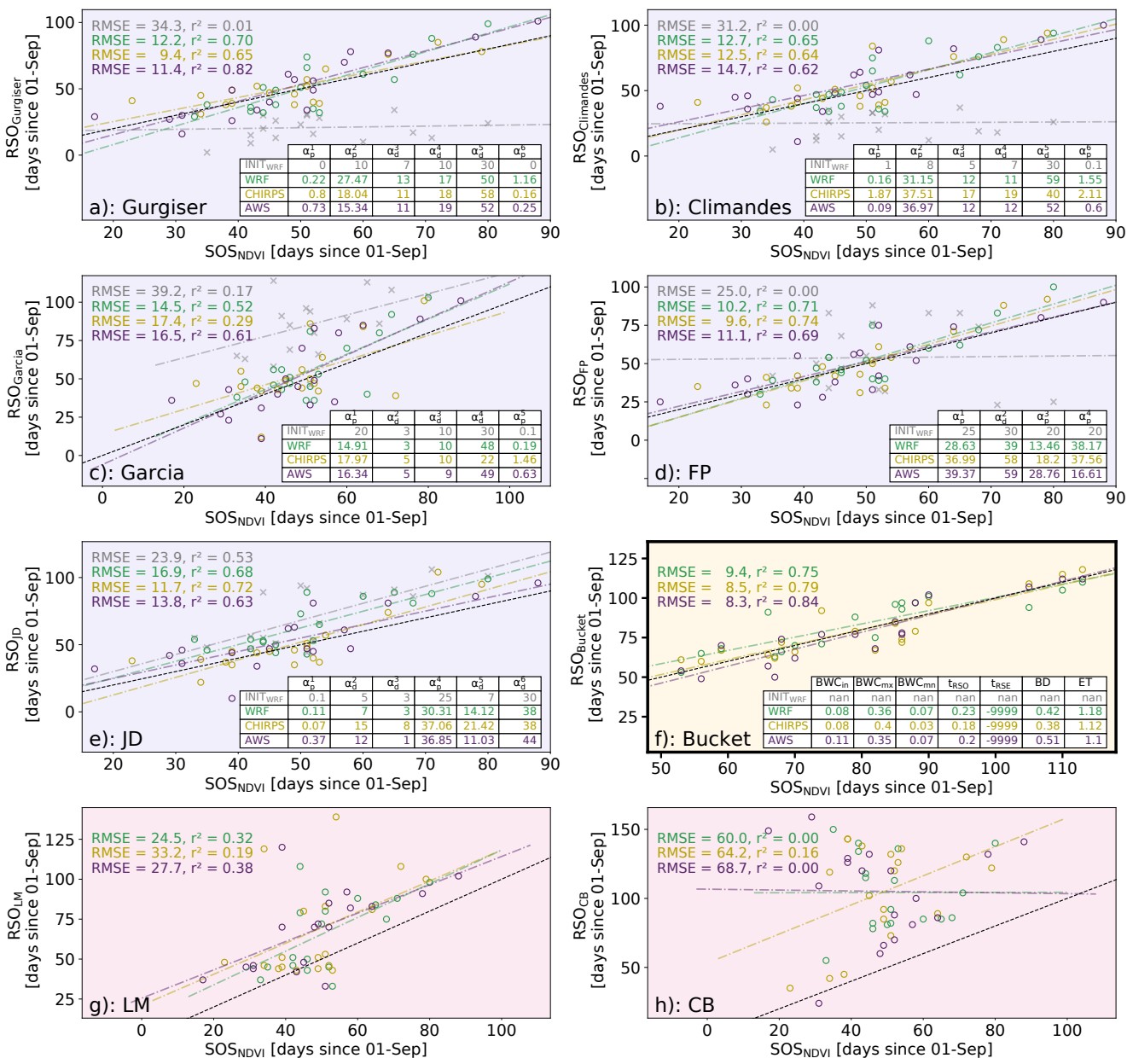

**Figure 2.** Rainy season onset metric outputs for WRF (teal), CHIRPS (orange) and AWS (purple) precipitation data. For the threshold-based metrics (purple background; panels a-e), results of the evaluation based on WRF but with uncalibrated thresholds as provided by the respective authors are also shown (gray) and denoted as $INIT_{WRF}$. The novel bucket methodology is highlighted in yellow (panel f) and the objective methods in red (panels g-h). The black dashed line indicates the 1:1 relationship with the $SOS_{NDVI}$, while the colored lines correspond to the regression of the parameters. Annotated are Root Mean Squared Error (RMSE) and the coefficient of determination ($r^2$). The tables correspond to the parameters as outlined in Table 1 and the equation in Section 2.3 after calibration for each of the precipitation data. The LM and CB method have no calibration and therefore no table is shown.

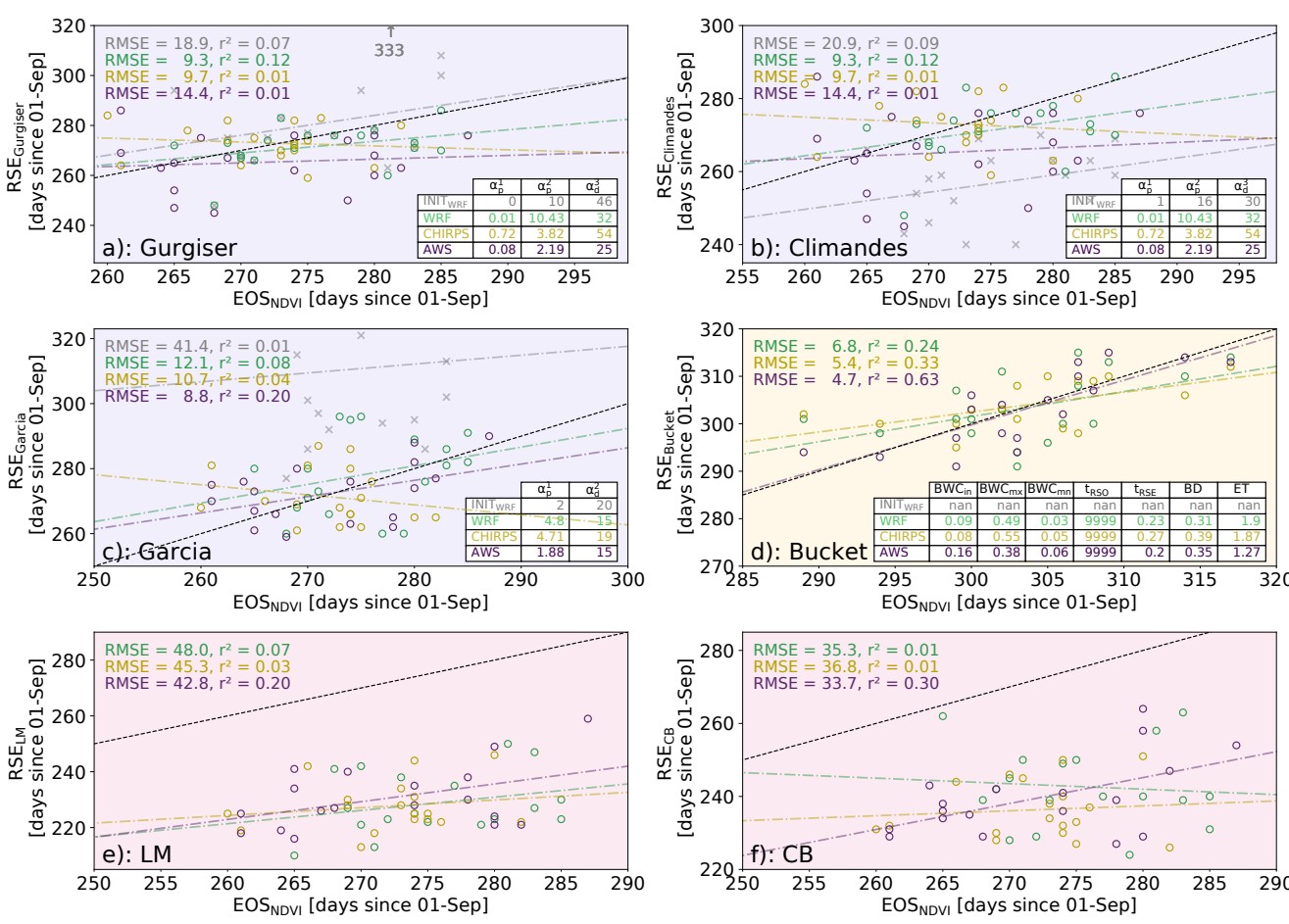

**Figure 3.** Same as Figure 2 but for Rainy Season End (RSE) and End of Season (EOS).

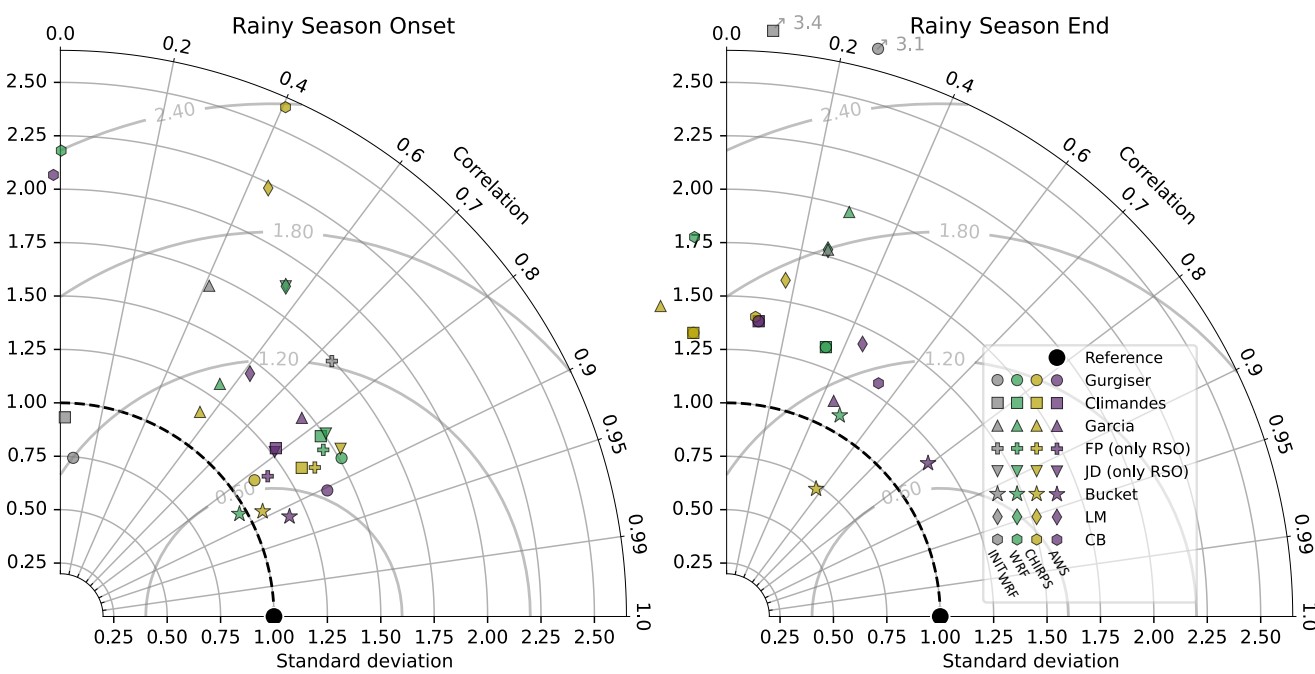

**Figure 4.** Taylor diagrams for normalized Rainy Season Onset (left) and End (right) for the calibration period 2000-2018. Each data point represents a metric depicted by the symbols, the colors represent the corresponding datasets (grey: WRF with initial metric parameters; teal: WRF calibrated; orange: CHIRPS; purple: AWS). The radial axes represent standard deviation, the azimuthal axis represents the correlation coefficient (r) and the circles the centered root mean squared difference. The black reference dot represents the normalized NDVI-derived SOS$_{NDVI}$ and EOS$_{NDVI}$ standard deviation.

## 3.2 Sensitivity analysis of rainy season metrics

To assess the sensitivity of rainy season metrics (RSO/RSE) to hydro-climatological changes, we correlated them with full hydrological year and sub-seasonal (SON, DJF, MAM, JJA) rainfall sums, as well as four ETCCDI climate indices as explained in Section 2.5. The results of these regressions are summarized in Figure 5, with detailed plots provided in Figures A4 and A5. In the context of the RSO (Fig. 5a, Fig. A4), all threshold-based metrics and the bucket metric show similar responses for both ETCCDI indices and precipitation sums, while LM and CB exhibit diverging sensitivities. Specifically, an increasing number

of dry days (DD) results in a later season onset, while a higher number of wet days (WD) leads to an earlier onset, with weaker correlations for LM and CB. With increasing heavy precipitation (R95pTOT), represented by the sum of precipitation falling above the 95th percentile relative to the control period (1980-2018), the correlation is negative for all threshold-based and the bucket metric, indicating a correlation between earlier rainy season onsets and more heavy precipitation. However, for LM, this relationship is positive, and for CB, the resulting slope is not significant. Similarly, an increase in average precipitation on

wet days, represented by the simple precipitation intensity index (SDII), results in an earlier season onset. LM again shows an opposite response, and CB shows just a weak correlation. All metrics except LM are sensitive to increased annual precipitation. With the exception of CB, which shows this correlation in DJF due its generally later onsets (c.f. Fig. 2), all metrics are strongly sensitive to SON precipitation. DJF, MAM, and partly JJA precipitation generally indicate this sensitivity as well, but this is most likely subject to autocorrelation. Notably, the bucket metric shows a stronger sensitivity to dry season (JJA) precipitation,

as its design of the metric allows for transferring information regarding water availability between hydrological years. Both LM and CB show a distinct positive correlation to increased MAM precipitation, indicating later rainy season onset. This is problematic because this correlation is a methodological artefact that does not reflect any physical process related to RSO water availability. This indicates limited metric robustness of the objective metrics to changes in the rainy season. Note that the start of CB is based on the period of the first 250 days of the rainy season, meaning that the metric is based only on information of

the period of September 1st to May 8th (March 19th to August 31st for the end).

Similar as previously seen in the calibration results (c.f. Figs. 3, 4b), the metrics for the end of the rainy season, RSE, show a weaker relationship with climate indices and precipitation sums, represented by lower $r^2$ values (see Fig. 5b and Fig. A5). For the number of dry (wet) days, all metrics suggest an earlier (later) season end, with the bucket model displaying the strongest

sensitivity. For both R95pTOT and SDII, most regressions are insignificant or show weak correlations, with the exception of the bucket model, which suggests a moderate correlation towards later rainy season ends, and CB suggesting the opposite. All metrics except CB suggest a moderate sensitivity to seasonal and total rainfall sums, with Garcia, and LM suggesting a negative slope for SON precipitation (for LM, also DJF). Gurgiser and Climandes suggest a very strong sensitivity to JJA rainfall. The RSE calculated by CB appears to be relatively insensitive to altered rainfall sums, being only significantly correlated to JJA

precipitation. Due to the lower overall correlation, interpreting these results is not as straightforward as for the RSO. However, the relatively high correlation of both the calibrated threshold-based and the bucket metric, along with revealing consistent correlations with our process understanding for all indices and precipitation sums, emphasizes their suitability for assessing

potential changes in water availability in semi-arid areas such as the Rio Santa basin.

Taken together, the sensitivity analysis reveals that for the RSO, all threshold-based models and the bucket model appear to produce appropriate results, while LM and CB are subject to sensitivities that are likely to hinder a reliable interpretation regarding the temporal manifestation of the rainy season, in particular when rainy season characteristics are expected to change. While less clear for the RSE, the overall message is similar with the bucket metric along with the threshold-based metrics being most reliable.

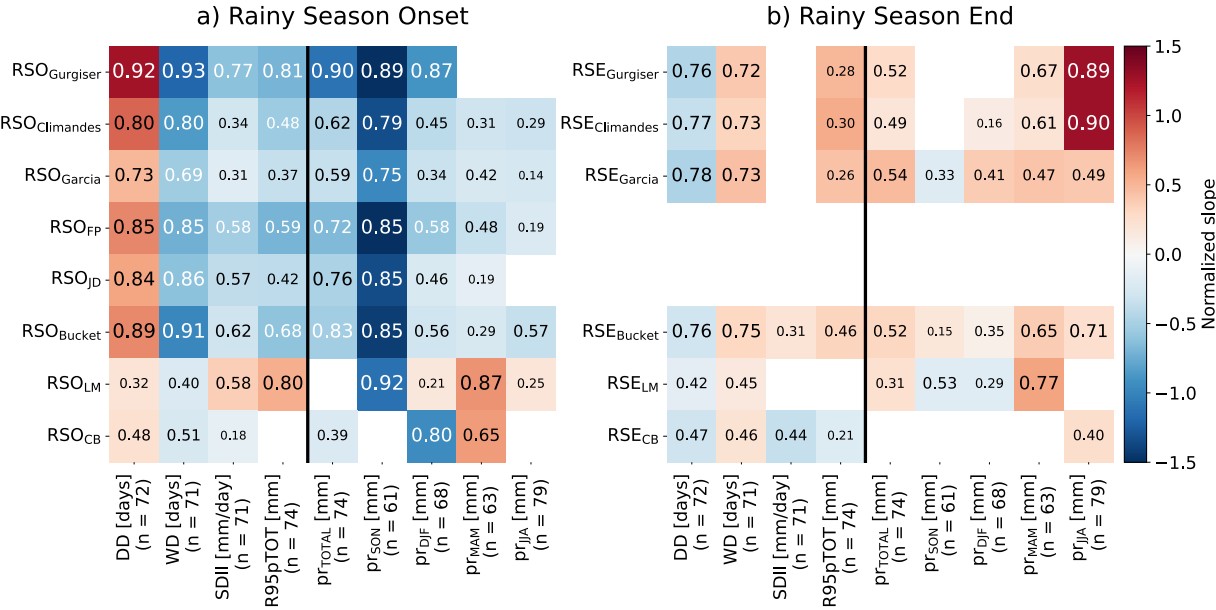

**Figure 5.** Heatmap of bin-weighted regression slopes with annotated $r^2$ values between ETCCDI indices and seasonal precipitation sums (independent variables) and rainy season metric derived onset and end (dependent variables). Corresponding bin sizes are noted on the x-axes labels as (n = x). Slope values are normalized and non-significant regressions (p > 0.01) are not shown. Full regressions including non-normalized slope values are displayed in Figs. A4 and A5.

## 3.3 Past & Future Trends

Finally, we calculated past metrics based on WRF data from 1981 to 2018 and projected future metrics up to 2100 using the statistically-downscaled CMIP5 model ensemble, which comprises 30 individual models (29 for RCP4.5), and subsequently evaluated the trends for the historical and the future period. As depicted in Fig. 6, the substantial variability observed in the RSO from 2000 to 2018 (average IQR over all 8 metrics = 16.4) seems to have existed similarly, or even more pronounced, in the preceding decades before 2000 in both time series (IQR = 27.0). Regarding the historical RSE, the missing data points in 1989/1990 in three metric outputs (Fig. 7a-c) are due to a dry spell lasting about three months leading to non-fulfilment of metric criteria and thus no-data labelling. Interestingly, LM and CB do not show any anomaly for this event because these metrics do not have information about any form of climatology. Conversely, this is accounted for by the bucket and threshold-based metric as the calibrated parameters represent the average climate of 2000 – 2018, such that extreme cases exceeding the calibration period cannot be informatively processed. We believe this is a desirable feature as for a practitioner this can be more informative than an unrealistic result in such cases. None of the metric outputs suggests a trend for the past period, either for the rainy season onset or the end of the rainy season (c.f. Figs. 6 and 7).

After establishing variability and trends for the historical period, we now explore the projected changes of rainy season metrics for the ensemble mean and standard deviation for each of the two RCP scenarios. Most of the metrics do not suggest a change in either the onset or end of the rainy season until the end of the century (Figs. 6 & 7). Only the JD and FP metrics suggest earlier rainy season onsets (approximately 0.5 days per decade earlier for the stabilization scenario, RCP4.5; Fig. 6). Meanwhile, only the bucket metric suggests a small delay in rainy season ends, with a decadal slope of approximately 0.35 to 0.6 days for both scenarios (Fig. 7). In light of the anticipated increase in future precipitation in the Rio Santa basin (5.8 % $\pm$ 6.3 %, RCP4.5 and 12.1 % $\pm$ 11.0 %, RCP8.5, Potter et al. (2023)), combined with the sensitivities of the metrics discussed in Section 3.2, the results appear surprising. To investigate the apparent contradiction between increasing future annual precipitation trends and little change in the onset or end of the rainy season, we apply a trend analysis across monthly precipitation sums for each month of the year in the future CMIP5 ensemble (Fig. A6). As this shows that the months September and October do not show significant trends for either scenario and for RCP4.5 only January and April show significant precipitation increases (c.f. Fig.A6), the annual results seem consistent. While the early-season months are highly relevant for the determination of the RSO, changes in the peak rainy season months are generally outside of the periods used by most of the metrics to determine start and end. In absolute values, these trends in the dry months are very small (with 0.046 for May, 0.022 for June and 0.004 mm/day decadal slopes for July and August, c.f. Fig. A6) while the calibrated values for the dry day threshold to determine RSE (c.f. Fig. 3 & Table 1) are in the order of 2-10 mm. Therefore, the absolute changes are likely too small to significantly alter the outputs of the threshold-based metrics. The consistent trends for both scenarios derived from the bucket metric stem from the fact that higher peak rainy season rainfall will keep the BWC at a higher level (c.f. Fig. A2) and the decrease in water availability and thus the resulting rainy season end will be delayed.

There is between-model variability in future predictions of both rainy season start and end for all metrics (Figs. A7 and A8) making the resulting trends debatable. This is represented by only 7 out of 30 RCP8.5 and 2 out of 29 RCP4.5 CMIP5 models suggesting a significant delay individually in case of the bucket metric, and one model even suggesting an earlier RSE under RCP4.5. An assessment of the distribution of significance of model trends for each metric and scenario can be found in Figs. A7 and A8. These results reflect observations and previous findings (e.g. Hänchen et al., 2022) regarding the larger variability in RSO compared to RSE as illustrated by the considerably smaller RSE standard deviation across all metrics (c.f. Figs. 6, 7).

The projections by Potter et al. (2023) we use are based on statistical downscaling of CMIP5 models. At the continental scale, many CMIP5 models were previously reported to poorly represent the South American Monsoon System (SAMS) (Bombardi and Carvalho, 2008), a challenge that is particularly pronounced in the topographically complex Andes. We compare our results to those of De la Cruz et al. (2025), who performed statistical downscaling based on meteorological stations in Peru using CMIP6 data and analyzed changes through the LM metric. De la Cruz et al. (2025) also project an increase in total precipitation, consistent with the findings of Potter et al. (2023), whose data informed our study. De la Cruz et al. (2025) also find no significant future changes in rainfall seasonality using the LM metric for the domain in which the Rio Santa basin is located. Furthermore, they highlight that GCMs have limited skill in simulating the interannual variability of rainy season onset and end, noting that many CMIP6 simulations still struggle to adequately represent the SAMS (see also Olmo et al., 2022). This suggests that the results from downscaled CMIP6 models and the downscaled CMIP5 models used in this study are consistent, at least based on the LM metric. Our findings contrast the results of Jones and Carvalho (2013), who used 6 CMIP5 models to predict future South American Monsoon System changes under an RCP 8.5 scenario on the continental scale and further suggest, using the LM metric, earlier rainy season onsets and later retreats. This could be related to several key differences which are the larger CMIP5 model ensemble used here, a spatial mismatch between the Rio Santa basin and the greater region, resolution differences or the fact that the LM metric can be subject to inconsistent sensitivities to hydroclimatic change as we previously showed (c.f. Fig. 5).

Future predictions are further complicated by the limited understanding of expected ENSO changes and its effects in the region. While Cai et al. (2023) recently suggested an increase in ENSO variability linked to anthropogenic climate change, reliable ENSO-related predictions about the potential alteration of the rainy season and general precipitation patterns in the Rio Santa basin specifically cannot confidently be made at this time. Our results incorporate a large number of calibrated and sensitivity-tested rainy season metrics, combined with a high-resolution, bias-corrected large-ensemble of future precipitation dataset. As such, we suggest that studies suggesting future change in rainy season timing should be interpreted with caution in terms of climate model ensemble robustness, and as our results indicate, critically reviewed towards the calibration of rainy season metrics.

Finally, as we are calibrating the metrics on a vegetation proxy, the effects of future increasing temperatures on evapotranspiration should also be considered, as these are expected increase in the Rio Santa basin with rising temperatures (see Potter

et al., 2023). This is likely to affect actual plant water availability and introduce uncertainty of currently unknown magnitude in the region. While this does not affect the rationales of the metrics, it will likely alter the applicability from a practitioner's perspective. In future endeavors, the bucket metric could be modified to accommodate this by altering the evapotranspiration parameter over time, which for our demonstrative purposes was set to a constant value. We decided against pursuing this adjustment for the future projections presented here because the bucket metric is not intended to replace the tasks of sophisticated hydrological models, and realistically estimating actual evapotranspiration in a data-sparse environment is a complex task in itself. Meanwhile, it is therefore crucial to consider that when metrics like these are applied with water users in mind, factors beyond precipitation change (i.e. rising temperatures, wet/dry-spell frequency) must also be taken into consideration to ensure their practical relevance.

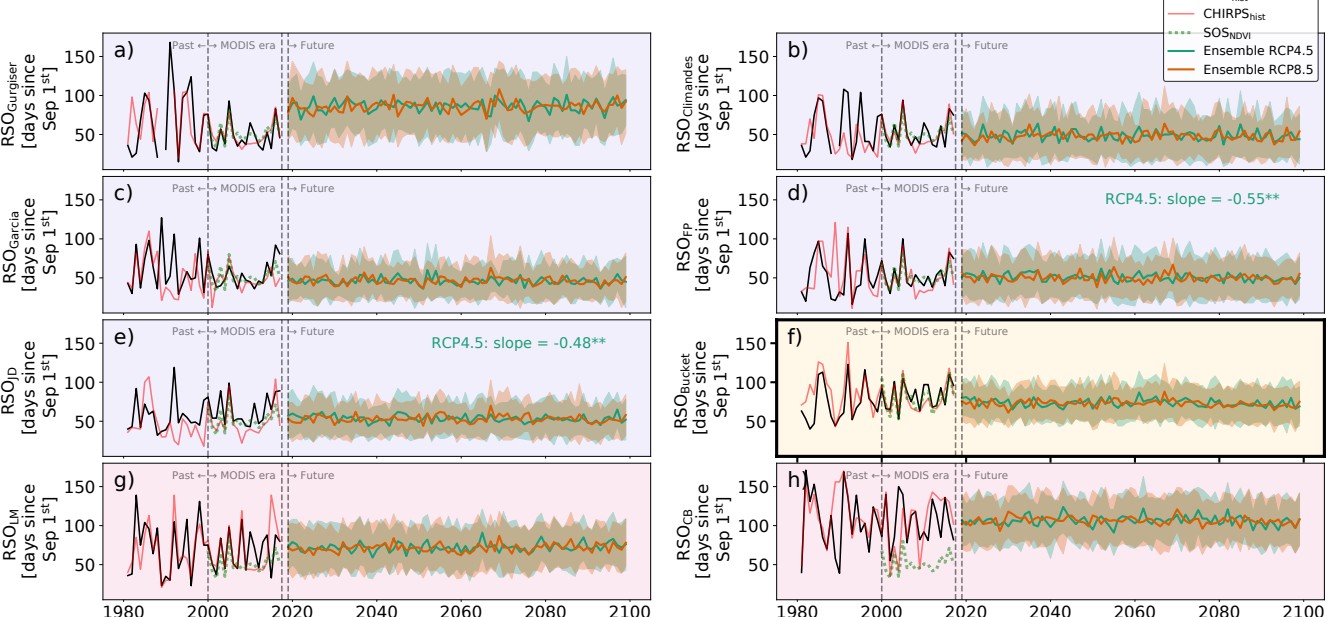

**Figure 6.** Rainy Season Onset (RSO) derived from 8 different metrics during the Past, Calibration (MODIS era), and Future periods, where threshold-based metrics are indicated by a purple, the bucket metric by a yellow and the objective metrics with a red background . Solid lines represent WRF (black) and CHIRPS (red) derived RSOs. The green line during the Calibration period indicates the $SOS_{NDVI}$ used for metric calibration. Teal (RCP4.5) and orange (RCP8.5) lines represent statistically downscaled CMIP5-model ensemble averages. Shading around these lines indicates one standard deviation from the mean across the statistically downscaled CMIP5 models. For WRF, CHIRPS, and the two CMIP5 ensembles, trends (denoted as days per decade) were derived through linear regression. Significant trends are denoted by asterisks: *** for $p < 0.01$ and ** for $p < 0.05$ while not significant trends ($p > 0.05$) are not displayed.

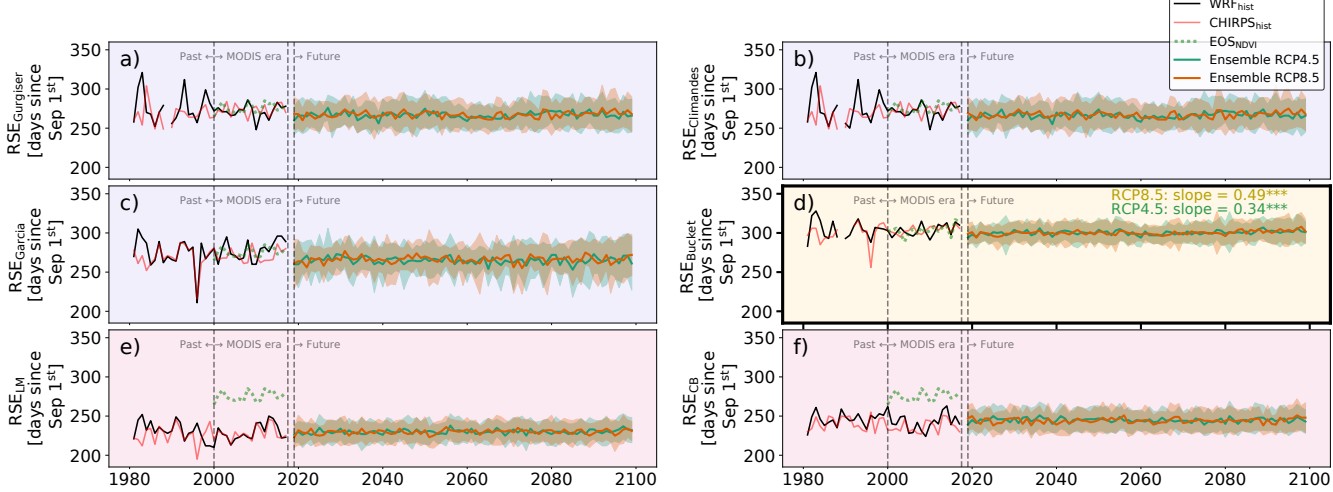

**Figure 7.** Same as Figure 6 but for Rainy Season End (RSE)

## 4    Conclusions

Based on several precipitation and remote-sensing derived land surface phenology data, we introduced a novel calibration
strategy for rainy season metrics applied in semi-arid regions. For all three considered precipitation datasets, we find that
the threshold-based rainy season metrics, once calibrated, are able to capture the interannual variability found in a vegetation
greenness proxy in the Rio Santa basin and exhibit sensible sensitivities to potential hydroclimatic changes. More objective
and flexible metrics on the other hand have comparably low skill regarding this task. These objective metrics seem to exhibit
implausible sensitivities that can potentially render them uninformative or even misleading under certain conditions of rainy
season change. We therefore recommend that the usage of such methods should be at least critically reviewed on a case-by-case
basis to ensure that no false conclusions are drawn or misleading practical recommendations are made.

Considering the numerous publications that highlight threshold-based metrics and propose a fixed parameter setup to be
suitable for specific regions, irrespective of the rainfall data source, we believe it is important to explore strategies for calibrat-
ing these metrics. This will enhance their practical application and effectiveness. Here, we demonstrated a framework for such
an approach using remotely sensed data on vegetation greenness. In the specific case of the Rio Santa basin, the vegetation
– rainfall correlation was proven reliable and, due to availability of NDVI data in relatively high spatial resolution, it is ideal
to resolve the complex terrain, where gridded rainfall products are often subject to resolution biases. We however do believe
that strategies for calibration different from using a proxy for vegetation greenness are also feasible as long as the variables are
correlated with rainfall inputs into the hydrological system and available in sufficient quality. Examples could be, but are not
limited to (undisturbed) runoff measurements or soil moisture data.

Motivated by limitations in existing metrics, we designed a novel bucket metric, which outperforms other metrics for both
the onset and end of the rainy season, shows physically consistent sensitivities and corrects for the vegetation – precipitation
lag. The high skill and flexibility of the bucket metric allows for a wide range of applications in the context of hydroclimate in
semi-arid areas. Additionally, it can likely be extended e.g. by making evapotranspiration dependent on energy- and/or water
availability or by altering parameters over time to simulate changes, while still remaining simplistic and efficient. The bucket
metric is to our knowledge also the first attempt to take legacy effects of water availability into account; particularly relevant
in regions such as the Rio Santa basin where large inter-annual precipitation anomalies, for example related to ENSO, are
common. Future attempts in addressing questions regarding the rainy season across semi-arid regions can readily use or adapt
the bucket metric to suit a wide range of requirements.

Using the bucket metric together with other calibrated and sensitivity-tested rainy season metrics and a large number of
future projections, we conclude that although precipitation is projected to increase, consistent trends for the rainy season
onset cannot be derived and we find a comparably small delay of the rainy season end and consequently an increase of the
rainy season length. Considering high regional inter-annual variability, large intermodel spread of the CMIP5 projections

and other factors currently poorly understood, such as the future impact of ENSO, reliable projections of climatic change in the tropical Andes remain challenging. While our novel framework allows crucial insights derived from rainfall time series, an adequate assessment of future water availability for practitioners' needs would benefit from more robust climate model forcings, eventually to be expected from the emergence of high-resolution convection-permitting model projections which will allow for better representation of local precipitation. In addition, evapotranspiration changes should be further investigated, most appropriately analyzed through a sophisticated eco-hydrological model. Until then, practitioners' as well as researchers can profit from more robust predictions of water availability building on our novel framework.

*Code and data availability.* Pre-processed data and python code to recreate the analysis and figures are available at https://github.com/lohae/RainySeasonMetrics and preserved at https://doi.org/10.5281/zenodo.13952139, allowing to apply and test the metrics for other regions or data. Full bias-corrected WRF data can be obtained at https://data.bas.ac.uk/full-record.php?id=GB/NERC/BAS/PDC/01728. The future precipitation from the statistically downscaled CMIP5 models are available at https://doi.org/10.5285/67CEB7C8-218C-46E1-9927-CFEF2DD95526, the future ETCCDI at https://doi.org/10.5285/B56D30E8-EDAA-4225-96D7-FCC689E930C7. Full CHIRPS data can be obtained through https://data.chc.ucsb.edu/products/CHIRPS-2.0/ while NDVI raw data can be acquired (for example) through https://appeears.earthdatacloud.nasa.gov/. The AWS data is publicly available at https://www.senamhi.gob.pe/?p=estaciones, we acquired it however through the METEODAT platform (available on request).

*Author contributions.* LH: Conceptualization, Data curation, Formal analysis, Investigation, Methodology, Software, Visualization, Writing – original draft preparation EP: Data curation, Formal analysis, Methodology, Writing – review & editing CK: Conceptualization, Data curation, Formal analysis, Software, Writing – review & editing PC: Conceptualization, Funding acquisition, Supervision, Writing – review & editing WG: Writing – review & editing FM: Data curation, Funding acquisition, Project administration, Resources, Supervision, Writing – review & editing GW: Conceptualization, Methodology, Resources, Software, Supervision, Writing – review & editing

*Competing interests.* The authors declare that they have no conflict of interest.

*Acknowledgements.* Parts of this study were conducted in the frame of the AgroClim Huaraz project, funded by the Earth System Sciences Program of the Austrian Academy of Sciences (OEAW) and the FFG project AustroSIF. EP acknowledges funding from a Leverhulme Trust Early Career Research Fellowship at the time of submission. CK also acknowledges funding from the NERC independent research fellowship COCOON (NE/X017419/1). We thank Mario Rohrer for providing access to the METEODAT platform where we acquired the weather station data. Special thanks to the developers of numpy (Harris et al., 2020), scipy (Virtanen et al., 2020), xarray (Hoyer and Hamman, 2017), pandas (McKinney), pwlf (Jekel and Venter, 2019), salem (Maussion et al., 2023) and their dependencies for making their code available on a free and open-source basis. Thanks to Yannick Copin for the taylor diagram code (available under https://gist.github.com/ycopin/3342888) and

495   to Santiago Belda and the IPL team for the DaTimeS software (available under https://artmotoolbox.com/plugins-standalone/91-plugins-standalone/34-datimes.html).

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

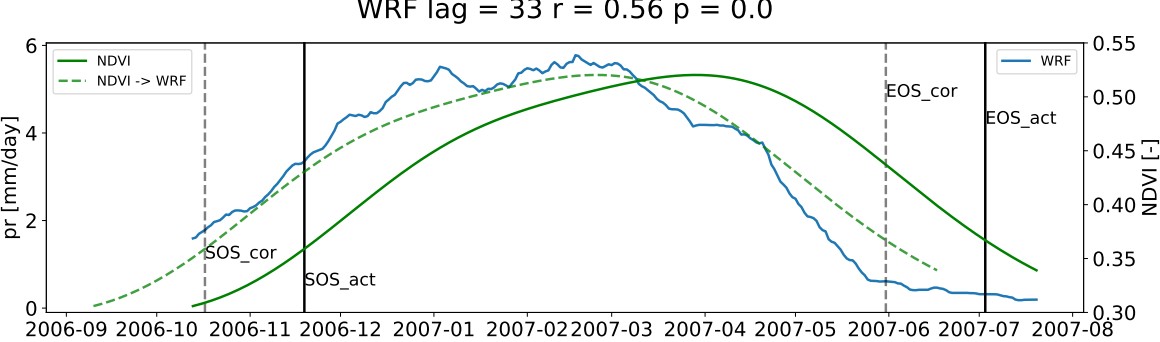

**Figure A1.** Visual example of the cross correlation function for lag correction of one hydrological year. The lag was determined based on WRF data smoothed by a 12-week rolling average and the processed NDVI data

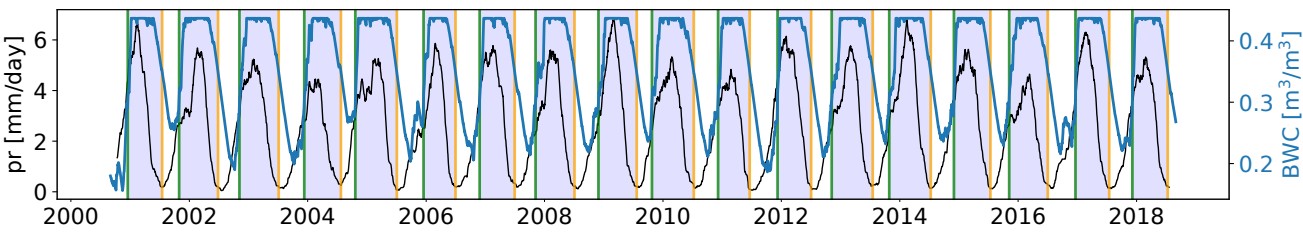

**Figure A2.** 12-week rolling window WRF time series (black) and BWC, modeled from daily (non-smoothed) precipitation from bucket metric (blue) for the calibration period 2000-2018. Green (orange) vertical lines indicate RSO (RSE) dates derived by the bucket metric. Blue shading indicates the resulting rainy season period

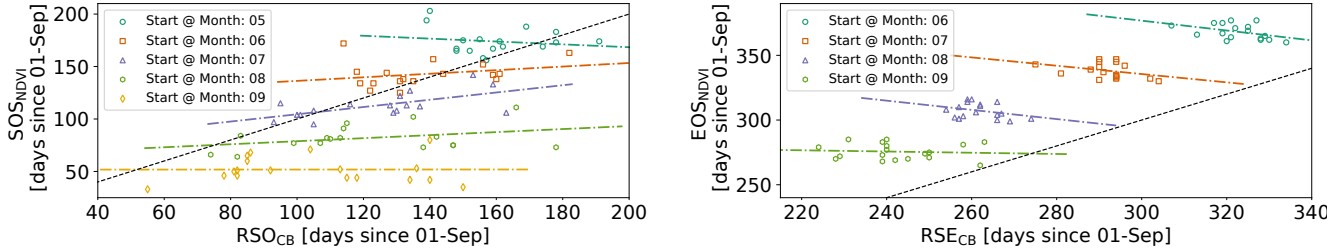

**Figure A3.** Sensitivity of Two-Phase linear Regression method to hydrological year definition by Cook and Buckley (2009).

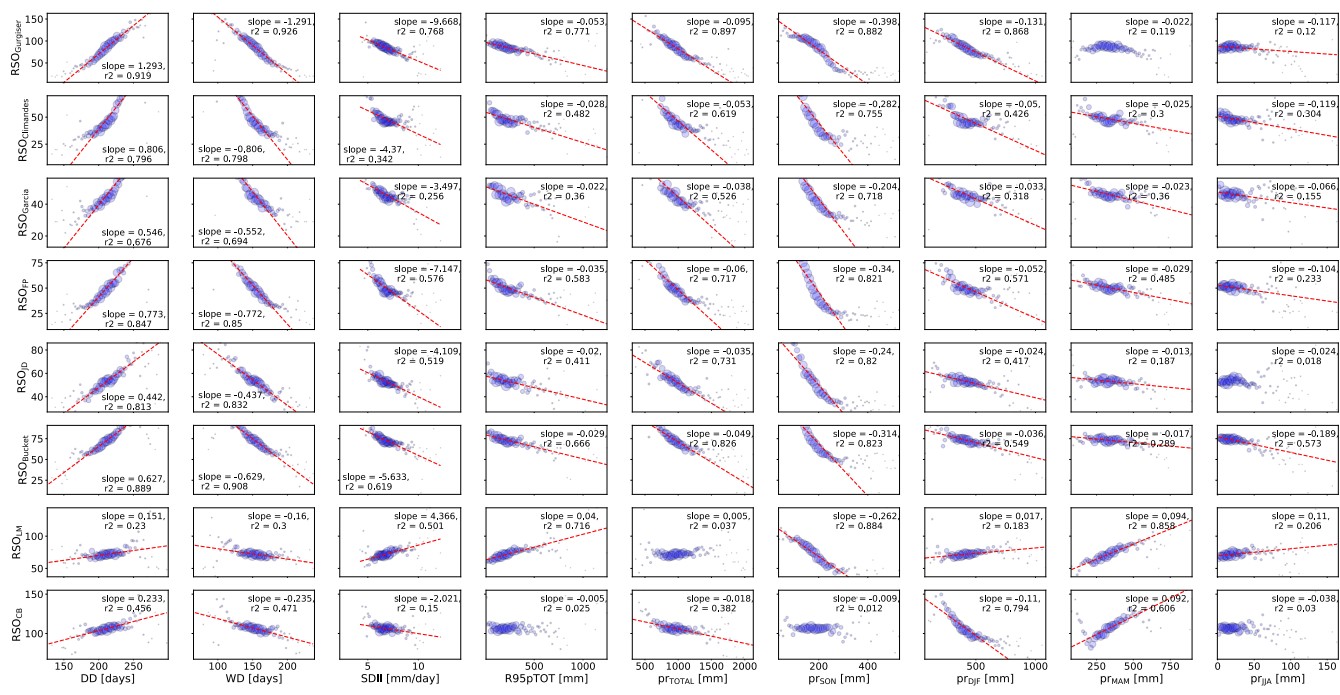

**Figure A4.** Bin-weighted regressions for RSO as summarized in 5a. Red regression lines are only shown for significant regressions (p < 0.01).

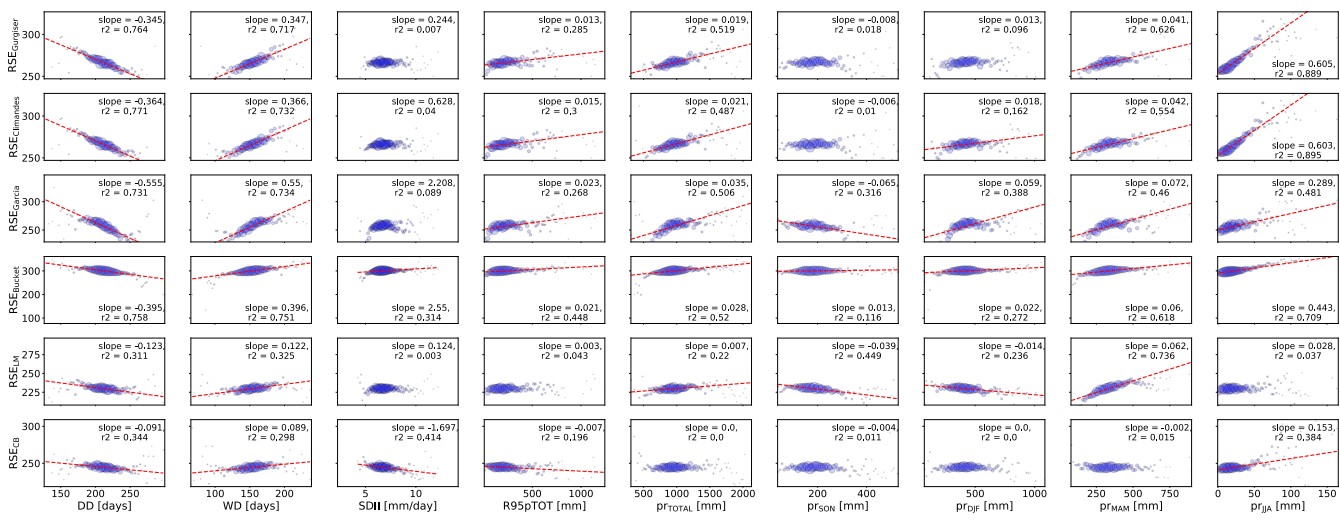

**Figure A5.** Bin-weighted regressions for RSE as summarized in 5b. Red regression lines are only shown for significant regressions (p < 0.01).

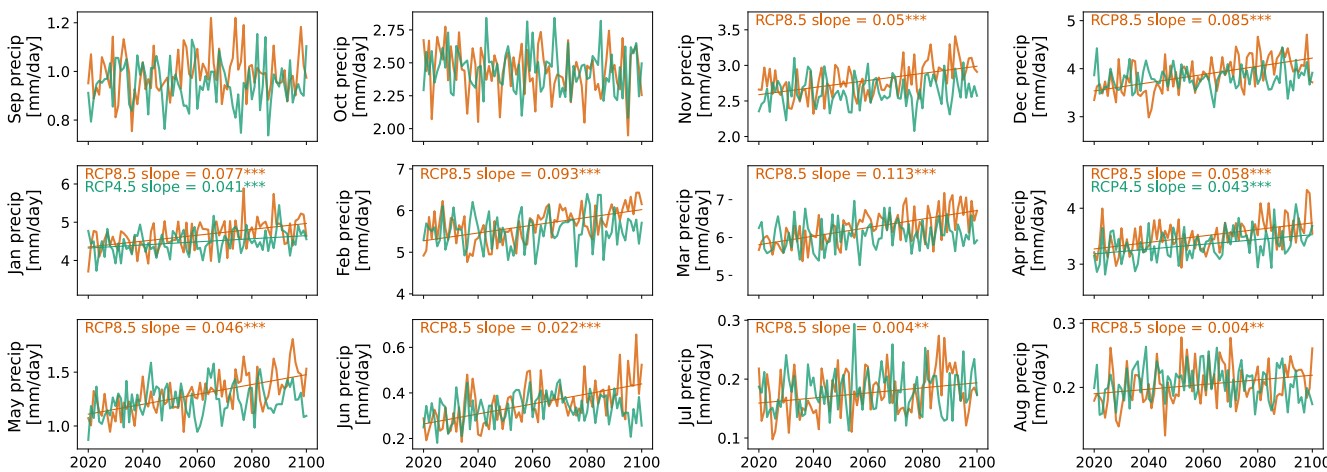

**Figure A6.** Monthly trends for the CMIP5 ensemble for both RCP4.5 (teal) and RCP8.5 (brown) scenarios. Decadal trends were derived through linear regression. Significant trends are denoted by asterisks: *** for p < 0.01 and ** for p < 0.05 while regression lines for non-significant trends (p > 0.05) are not displayed.

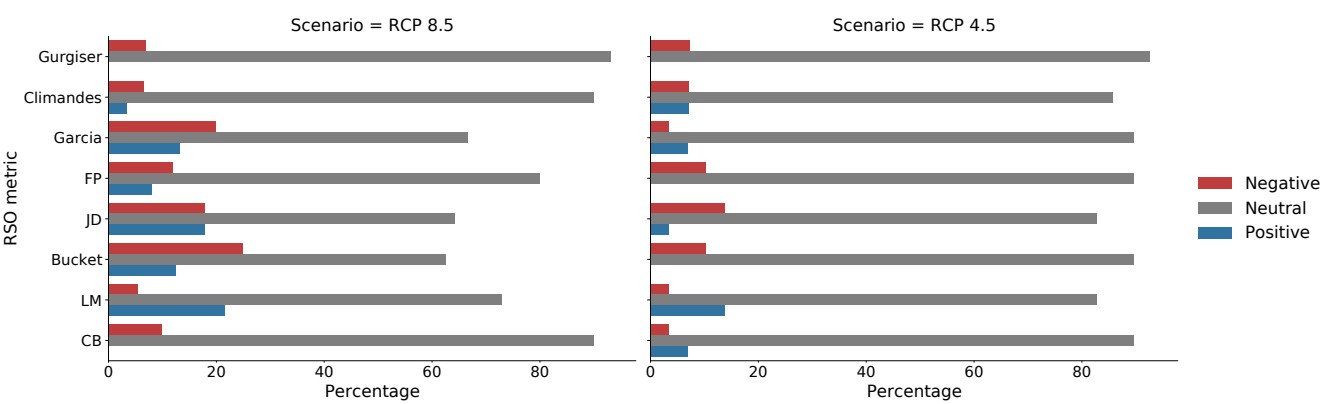

**Figure A7.** Relative Distribution of significant and non-significant CMIP5 model timeseries (p < 0.05) and their sign for the derived rainy season onsets for the timeperiod 2019-2100 for each rainy season metric and both RCP scenarios. A negative trend refers to an earlier season start and a positive trend to a later start.

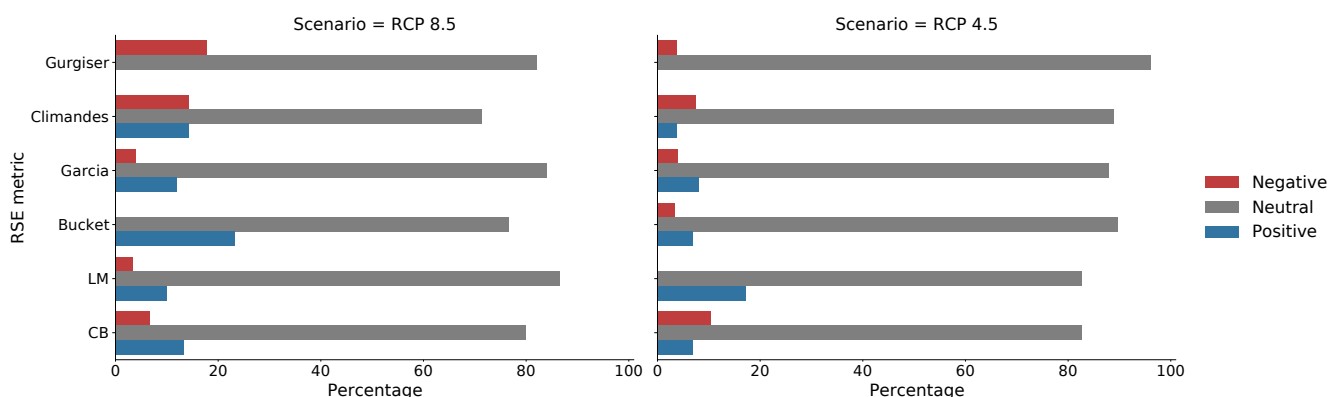

**Figure A8.** Same as Figure S8 but for RSE.