# Peer review of "A Novel Framework for Analyzing Rainy Season Dynamics in semi-arid environments: A case study in the Peruvian Rio Santa Basin"

_EGUsphere, 2024_

## Author Response (AR1)

This manuscript presents a case study in understanding rainy season dynamics in semi-arid regions. The study is intended to develop and validate a new framework for calibrating rainy season metrics. The paper is generally clear.

Dear Jingwei Zhou, thanks a lot for taking the time to review our research and for your thoughtful comments allowing us to significantly improve our manuscript. Please find our point by point responses below. Note that all line number references are referring to the cleaned revised manuscript and not the version with track changes. For your orientation, we use **bold** letters to indicate new or revised text and *italic* letters to indicate citations.

**Comment #1.1**

Although CMIP5 data seem to be applicable in your research, I will still suggest you using the newer CMIP6 data, which are accessible in many data portals provided by organizations such as ESGF Copernicus. CMIP5 data has been more than ten years since developed, while CMIP6 incorporate numerous updates and incorporate more features, let alone it may address the issues you do mentioned in your text (lines 345-350). In addition, the implications or significance for this novel method in evaluating rainy season and to be applied in future research could be discussed further. A brief comment on the potential transferability of this framework to other semi-arid regions would be valuable as well. Overall, I will give this paper the suggestion of moderate revisions

Thanks for your suggestions.

Regarding CMIP5 vs. CMIP6: Please note that we do not directly use CMIP5 data but rather use a high resolution statistically downscaled CMIP5 data product which was specifically created for the area by Potter et al. (2023). This dataset is the only existing high-resolution, gridded future dataset which combines a large ensemble of future climate models for the Rio Santa basin. While we agree that a dataset based on CMIP6 would be interesting, it however simply does not exist at this point. Given that the main focus of this paper is to compare, validate and improve upon metrics to determine the onset and end of the rainy season, we believe including the creation of a new future climate dataset would obscure the main purpose of this manuscript.

Using statistically-downscaled CMIP5 models based on a bias-corrected regional climate model has advantages over downscaling directly to station data, as the regional climate model is spatially and temporally consistent, without the potential biases which come from using station data (for example there being fewer stations at higher elevations due to inaccessibility). However even using statistical rather than dynamical downscaling methods, creating large-ensembles of high-resolution gridded datasets is computationally expensive. As such, it is not feasible to redo previously-published work for this study.

To this end, we already stated in the preprint that the climate data we use in our analysis is unique and comes from a previously published study:

Introduction, l.105f.:

"*Specifically, we calculate the rainy season metrics based on convection-permitting, bias-corrected Weather Research and Forecasting (WRF) precipitation data and statistically downscaled CMIP5 projections (Potter et al., 2023) [...]*"

We revised the corresponding part in the data section of the manuscript to emphasize our rationale for choosing these data:

Methods/Data, l.145:

**A key component is the WRF bias-corrected regional climate model data published by Potter et al. (2023), which provides consistent precipitation estimates at 4 km grid spacing from 1981 to 2018. In addition, Potter et al. (2023) produced statistically downscaled projections based on a 30-member CMIP5 ensemble from 2019 to 2100 using quantile delta mapping for both the RCP4.5 and RCP8.5 scenarios. These data preserve CMIP5 model trends while adjusting precipitation magnitude and the number of wet days based on the bias-corrected WRF data, and are available in the same 4 km grid spacing from 2019 to 2100. The two RCP scenarios allow us to assess multiple trajectories of future changes in the rainy season in the Rio Santa basin and provide a large dataset for metric sensitivity analysis. We do not evaluate metrics for raw, coarse-scale CMIP data in this study, as at their native resolution, they are known to inadequately represent orographic processes and interannual variability (e.g. Gutierrez et al., 2024).**

In the section "Sensitivity analysis" we stated that *"[...] we used four Expert Team on Climate Change Detection and Indices (ETCCDI) climate indices (Zhang et al., 2011) based on the WRF and CMIP5 data by Potter et al., (2023)."* We acknowledge that this might read as if we created some data here - but in fact these data were first published in the previous work by Potter et al., (2023). To avoid this confusion, we changed this sentence to, l.239:

*"[...] we used four Expert Team on Climate Change Detection and Indices (ETCCDI) climate indices (Zhang et al., 2011) based on the WRF and* **statistically downscaled** *CMIP5 data* **created by** *Potter et al., (2023)."*

Finally, to ensure that there are no misunderstandings, we edited a sentence of the Introduction, specifically stating that we make use of results and data of previous studies, l.104:

**"[...] we employ a multi-faceted approach capitalizing on previous studies: We combine several precipitation datasets with remote sensing data on temporal vegetation development."**

We want to state that we take your comment seriously. Therefore, we have added a comparison to a study (published only after we submitted our original manuscript) by De la Cruz et al. (2025). The authors used the Liebmann metric to determine future changes to rainy season timings over a network of weather stations in Peru. Both that study, which used a statistical downscaling approach of CMIP6 data onto in-situ precipitation observations, and this study

based on statistically down-scaled CMIP5 data using a convection-permitting model, find no statistically significant future changes of the onset or end of the rainy season using the Liebmann method. This suggests that conclusions on future changes of rainy season timings are likely to be robust with respect to the input climate dataset (CMIP5 and CMIP6), at least for the Liebmann metric. We revised our introduction now referring to their work:

Introduction, l.86:

**"Potential shifts in the timing of the rainy season in the region, despite their profound implications for both societal and ecological systems, have only recently been assessed. Notably, De la Cruz et al. (2025) used an objective metric to derive rainfall sums and rainy season onset and end for a Peru-wide network of meteorological stations based on statistically downscaled CMIP6 projections to derive future changes. They found an increase in future annual precipitation and similar to other studies show that past rainy season dynamics in the broader Andean region reveal high inter-annual variability in rainy season onset, with generally non-significant or weak longer-term trends (Garcia et al., 2007; Giráldez et al., 2020; Gurgiser et al., 2016; Sedlmeier et al.,2023)."**

The same authors suggest in their discussion that *"GCMs showed more favourable results for accumulated rainfall, but had limitations in simulating the onset and cessation, with correlations ranging from 0.2 to 0.6. Studies in other regions have demonstrated more optimal results for models with higher resolution."* pointing towards RCMs being the way forward rather than GCM ensembles. We added to the manuscript that we find similar results for the Liebmann method for the future period and make the argument that the data by Potter et al. (2023), based on CMIP5, are thus not compromising our results.

Results & Discussion, Past & Future Trends, l.398:

**"The projections by Potter et al. (2023) we use are based on statistical downscaling of CMIP5 models. At the continental scale, many CMIP5 models were previously reported to poorly represent the South American Monsoon System (SAMS) (Bombardi and Carvalho, 2008), a challenge that is particularly pronounced in the topographically complex Andes. We compare our results to those of De la Cruz et al. (2025), who performed statistical downscaling based on meteorological stations in Peru using CMIP6 data and analyzed changes through the LM metric. De la Cruz et al. (2025) also project an increase in total precipitation, consistent with the findings of Potter et al. (2023), whose data informed this study. De la Cruz et al. (2025) also find no significant future changes in rainfall seasonality using the LM metric for the domain in which the Rio Santa basin is located. Furthermore, they highlight that GCMs have limited skill in simulating the interannual variability of rainy season onset and end, noting that many CMIP6 simulations still struggle to adequately represent the SAMS (see also Olmo et al., 2022). This suggests that the results from downscaled CMIP6 models and the downscaled CMIP5 models used in this study are consistent, at least based on the LM metric."**

Regarding your comments on further discussion of applying our framework in future research and potential transferability, there are several important aspects to consider:

1. The widely adapted objective methods which showcase implausible sensitivities should be used with caution. It seems that the general opinion in the community is that these metrics are robust to changes as they do not depend on specific parameters. However, in our sensitivity analysis we show that this is in fact not the case. We now added a sentence further emphasizing this argument into the conclusions, l.428: *"[...] while more objective and flexible metrics have comparably low skill regarding this task. These objective metrics seem to exhibit implausible sensitivities that can potentially render them uninformative or even misleading under certain conditions of rainy season change. "***We therefore recommend that the usage of such methods should be at least critically reviewed on a case-by-case basis to ensure that no false conclusions are drawn or misleading practical recommendations are made."**

2. We show that our calibration approach is effective and transferable and should be considered by future research. Among other sections in the manuscript where we mention that the approach can be readily adapted, most prominently we revised the abstract following a comment of Reviewer #2 (Comment #2.2), where we now state, l.20: **"Our findings emphasize the need for careful calibration of metrics across diverse climate scenarios and different locations to ensure their reliability for agricultural planning, policymaking, and climate adaptation strategies"**

3. We show that our bucket approach is outperforming other metrics and can be further developed or adapted depending on the specific question. We already dedicated a paragraph of the conclusions to it, which we now revised to further acknowledge this argument, l.444: **"Motivated by limitations in existing metrics, we designed a novel bucket metric, which outperforms other metrics for both the onset and end of the rainy season, shows physically consistent sensitivities and corrects for the vegetation – precipitation lag. The high skill and flexibility of the bucket metric allows for a wide range of applications in the context of hydroclimate in semi-arid areas. Additionally, it can likely be extended e.g. by making evapotranspiration dependent on energy- and/or water availability or by altering parameters over time to simulate changes, while still remaining simplistic and efficient. The bucket metric is to our knowledge also the first attempt to take legacy effects of water availability into account; particularly relevant in regions such as the Rio Santa basin where large inter-annual precipitation anomalies, for example related to ENSO, are common. Future attempts in addressing questions regarding the rainy season across semi-arid regions can readily use or adapt the bucket metric to suit a wide range of requirements"**

4. Finally, while not specifically mentioned in the text, we would like to point out that all the code used for the analysis and the preprocessed data is available in a public repository (https://github.com/lohae/RainySeasonMetrics) and preserved at Zenodo (https://doi.org/10.5281/zenodo.13952139), allowing other researchers to build upon it.

5. Together with answering a comment by reviewer #2 (comment #2.8) we now state in our objective statement more specifically the issue of transferability, l.XY: **"The principal**

**objective of this study is to showcase a novel framework for characterizing the rainy season, emphasizing the importance of employing a calibration strategy for inferred rainy season onsets and ends. In addition, we test the sensitivity of rainy season metrics to plausible changes in rainfall intensity and frequency, as might occur due to global warming. By capturing shifts in seasonal rainfall dynamics, our approach provides a foundation for identifying and understanding hydrological changes that may inform future adaptation strategies. The proposed framework is designed to improve our understanding of variations in water availability within semi-arid regions, offering insights that extend beyond the Rio Santa basin and can be applied to similar climates. Regarding the Rio Santa basin, we aim to provide insights into past and future changes. We achieve this by: [...]"**

Specific Comments:

**Comment #1.2**

Line 17: I suggest change it to "only a slight delay in rainy season end, but no consistent trends in rainy season onset," to stress the later parts with the bad results

Thank you. Following your suggestion while still appropriately integrating the latter part of the sentence ("inter-annual variability and ensemble spread being the dominant factors"), we have rephrased the sentence as follows, see l.18:

**"Statistically downscaled CMIP5 ensemble projections for the future period suggest only a slight delay in the rainy season end, with no consistent trends in onset timing. Instead, inter-annual variability and ensemble spread remain the dominant influences."**

**Comment #1.3**

Lines 40-45: The review of existing metrics could be more comprehensive, particularly regarding their applications in similar geographical contexts. Some results from the former study can be put here.

Thank you. Following your comment #1.1 regarding CMIP5/CMIP6 and the fact that the paper by De la Cruz et al., (2024) was published after we submitted our manuscript, we updated that part, now to more thoroughly discuss results of usage of metrics in the broader region. Please refer to the edits in our answer to your comment #1.1

**Comment #1.4**

Line 123: maybe you can mention why you choose these two scenarios

Thank you. We are limited to these two scenarios due to the available data published by Potter et al. (2023), see also our response to your comment #1.1 which led to rewriting this part. Specifically related to this comment, we now state in l.150: "**The two RCP scenarios allow us**

**to assess contrasting trajectories of future changes in the rainy season in the Rio Santa basin and allow us to have a large data basis for metric sensitivity analysis."**

**Comment #1.5**

Lines 254-269: I think most of these can be moved into methods, just leave some brief descriptions here in the results

Thanks for your suggestion. We have made a new subsection in the methods section where we moved this part. Note that we initially decided to put this method-heavy part in the results section as most of these methods are rather standard repertoire (sums, linear regression…) but thought it would be good if readers did not have to jump back to the methods section here. We however believe both versions to work and decided to follow your suggestion. Please refer to the new section 2.5 for the part we moved into the methods section. The original paragraph in Section 3.2 is now shortened to, at l.308:

**To assess the sensitivity of rainy season metrics (RSO/RSE) to hydro-climatological changes, we correlated them with full hydrological year and sub-seasonal (SON, DJF, MAM, JJA) rainfall sums, as well as four ETCCDI climate indices as explained in Section 2.4. The results of these regressions are summarized in Figure 5, with detailed plots provided in Figures A4 and A5.**

**Comment #1.6**

Line 320: "Feb."

Now l.205 and changed to: **"February"**, for consistency throughout the manuscript we did that as well in l. 161.

**Comment #1.7**

Lines 345-367: Most of these can be moved into conclusions part. They are discussions from my perspective

Thanks for your suggestion. Note that we generally combined Results and Discussion here (i.e. Section 3 Results & Discussion), therefore the 3 subsections (~ 3.1 Comparison, 3.2 Metric sensitivity and 3.3 Past/Future Trends) all contain parts which are discussion. We decided to do that as we realized when reviewing an earlier version of the manuscript among the co-authors that the paper is otherwise harder to follow. The conclusions' part on the other hand is strictly summarizing our main findings and giving a broader outlook. We hope you understand that we would therefore prefer not to move this quite specific part related to limitations into the conclusions part.

**Comment #1.8**

Figure 1: you could move the index plot of Rio Santa Basin with NDVI a little further away from the basic topography map. The Basin looks like it's a big part located within the eastern part of the map

We adjusted the position of the inset map so that the limits of both the x and y axes extend beyond the boundaries of the larger map. This ensures it is clear that the basin does not lie within the larger map. Additionally, we changed the perimeter color to red to draw the viewer's attention to the inset map. We acknowledge that another version like 2 independent panels of similar size might be another solution but as this would heavily increase the figure size, the readability of other features of the map would be reduced. We hope we found a satisfactory compromise with the new version.

**New Figure 1:**

[Figure]

Figures 2 and 3: you could add a legend indicating different data sources and also different methodsdon't

Thanks, following your recommendation we critically revisited the figure but realized that the information is already there. The table does contain the color-coding of the data source. A legend would just repeat this information and make the figure unnecessarily busy.

So far each panel was one different rainy season metric, however indicated only by the relatively small y-label, per category by the background color and indexed by a),b)....h). As we agree that this information was not very easy to find in such a busy figure, we now added the name of each metric to the letters.

**New Figure 2:**

[Figure]

**New Figure 3:**

[Figure]

**Comment #1.10**

Figures 6 and 7: maybe add a legend showing different data sources and different scenarios

Thanks, we agree, the figure was indeed missing a legend. We added a legend to both of the figures according to your suggestions.

New Figures 6 & 7:

[Figure]

**Comment #1.11**

Some paragraphs have indentation while others , please keep them consistent

Apologies for the oversight, the revised manuscript has now consistent indentation.

The manuscript presents an interesting and well-structured approach to understanding rainy season dynamics in semi-arid regions. It contributes meaningfully to hydrology and earth system sciences by introducing a novel framework for calibrating rainy season metrics with vegetation data, addressing their applicability under current and future climate scenarios. However, there are areas where improvements can enhance the manuscript's scientific rigor and presentation quality.

Thank you for taking the time reviewing our manuscript and for your constructive and thoughtful suggestions which helped us to improve our manuscript. Please find our point-by-point responses below. Note that all line numbers references are referring to the revised manuscript and not the version with track changes. For your orientation, we use **bold** letters to indicate new or revised text and *italic* letters to indicate citations.

**Comment #2.1**

While the manuscript provides valuable insights using CMIP5 data, incorporating CMIP6 would likely enhance the study due to historical forcings, and more comprehensive future scenarios. As noted by another reviewer, it would be appropriate to perform the study with CMIP6 to align with current advancements in climate modeling. Otherwise, the authors should provide a solid justification for relying on CMIP5, addressing how its limitations might affect the results and conclusions of the study.

Thank you. As you already pointed out, the other reviewer also made a similar comment. We therefore took the liberty to copy our comprehensive answer to the other reviewer here:

Regarding CMIP5 vs. CMIP6: Please note that we do not directly use CMIP5 data but rather use a high resolution statistically downscaled CMIP5 data product which was specifically created for the area by Potter et al. (2023). This dataset is the only existing high-resolution, gridded future dataset which combines a large ensemble of future climate models for the Rio Santa basin. While we agree that a dataset based on CMIP6 would be interesting, it however simply does not exist at this point. Given that the main focus of this paper is to compare, validate and improve upon metrics to determine the onset and end of the rainy season, we believe including the creation of a new future climate dataset would obscure the main purpose of this manuscript.

Using statistically-downscaled CMIP5 models based on a bias-corrected regional climate model has advantages over downscaling directly to station data, as the regional climate model is spatially and temporally consistent, without the potential biases which come from using station data (for example there being fewer stations at higher elevations due to inaccessibility). However even using statistical rather than dynamical downscaling methods, creating large-ensembles of high-resolution gridded datasets is computationally expensive. As such, it is not feasible to redo previously-published work for this study.

To this end, we already stated in the preprint that the climate data we use in our analysis is unique and comes from a previously published study:

Introduction, l.105f.:

"*Specifically, we calculate the rainy season metrics based on convection-permitting, bias-corrected Weather Research and Forecasting (WRF) precipitation data and statistically downscaled CMIP5 projections (Potter et al., 2023) [...]*"

We revised the corresponding part in the data section of the manuscript to emphasize our rationale for choosing these data:

Methods/Data, l.145:

**A key component is the WRF bias-corrected regional climate model data published by Potter et al. (2023), which provides consistent precipitation estimates at 4 km grid spacing from 1981 to 2018. In addition, Potter et al. (2023) produced statistically downscaled projections based on a 30-member CMIP5 ensemble from 2019 to 2100 using quantile delta mapping for both the RCP4.5 and RCP8.5 scenarios. These data preserve CMIP5 model trends while adjusting precipitation magnitude and the number of wet days based on the bias-corrected WRF data, and are available in the same 4 km grid spacing from 2019 to 2100. The two RCP scenarios allow us to assess multiple trajectories of future changes in the rainy season in the Rio Santa basin and provide a large dataset for metric sensitivity analysis. We do not evaluate metrics for raw, coarse-scale CMIP data in this study, as at their native resolution, they are known to inadequately represent orographic processes and interannual variability (e.g. Gutierrez et al., 2024).**

In the section "Sensitivity analysis" we stated that *"[...] we used four Expert Team on Climate Change Detection and Indices (ETCCDI) climate indices (Zhang et al., 2011) based on the WRF and CMIP5 data by Potter et al., (2023)."* We acknowledge that this might read as if we created some data here - but in fact these data were first published in the previous work by Potter et al., (2023). To avoid this confusion, we changed this sentence to, l.239:

*"[...] we used four Expert Team on Climate Change Detection and Indices (ETCCDI) climate indices (Zhang et al., 2011) based on the WRF and* **statistically downscaled** *CMIP5 data* **created by** *Potter et al., (2023)."*

Finally, to ensure that there are no misunderstandings, we edited a sentence of the Introduction, specifically stating that we make use of results and data of previous studies, l.104:

**"[...] we employ a multi-faceted approach capitalizing on previous studies: We combine several precipitation datasets with remote sensing data on temporal vegetation development."**

We want to state that we take your comment seriously. Therefore, we have added a comparison to a study (published only after we submitted our original manuscript) by De la Cruz et al. (2025). The authors used the Liebmann metric to determine future changes to rainy season timings over a network of weather stations in Peru. Both that study, which used a statistical downscaling approach of CMIP6 data onto in-situ precipitation observations, and this study

based on statistically down-scaled CMIP5 data using a convection-permitting model, find no statistically significant future changes of the onset or end of the rainy season using the Liebmann method. This suggests that conclusions on future changes of rainy season timings are likely to be robust with respect to the input climate dataset (CMIP5 and CMIP6), at least for the Liebmann metric. We revised our introduction now referring to their work:

Introduction, l.86:

**"Potential shifts in the timing of the rainy season in the region, despite their profound implications for both societal and ecological systems, have only recently been assessed. Notably, De la Cruz et al. (2025) used an objective metric to derive rainfall sums and rainy season onset and end for a Peru-wide network of meteorological stations based on statistically downscaled CMIP6 projections to derive future changes. They found an increase in future annual precipitation and similar to other studies show that past rainy season dynamics in the broader Andean region reveal high inter-annual variability in rainy season onset, with generally non-significant or weak longer-term trends (Garcia et al., 2007; Giráldez et al., 2020; Gurgiser et al., 2016; Sedlmeier et al.,2023)."**

The same authors suggest in their discussion that *"GCMs showed more favourable results for accumulated rainfall, but had limitations in simulating the onset and cessation, with correlations ranging from 0.2 to 0.6. Studies in other regions have demonstrated more optimal results for models with higher resolution."* pointing towards RCMs being the way forward rather than GCM ensembles. We added to the manuscript that we find similar results for the Liebmann method for the future period and make the argument that the data by Potter et al. (2023), based on CMIP5, are thus not compromising our results.

Results & Discussion, Past & Future Trends, l.398:

**"The projections by Potter et al. (2023) we use are based on statistical downscaling of CMIP5 models. At the continental scale, many CMIP5 models were previously reported to poorly represent the South American Monsoon System (SAMS) (Bombardi and Carvalho, 2008), a challenge that is particularly pronounced in the topographically complex Andes. We compare our results to those of De la Cruz et al. (2025), who performed statistical downscaling based on meteorological stations in Peru using CMIP6 data and analyzed changes through the LM metric. De la Cruz et al. (2025) also project an increase in total precipitation, consistent with the findings of Potter et al. (2023), whose data informed this study. De la Cruz et al. (2025) also find no significant future changes in rainfall seasonality using the LM metric for the domain in which the Rio Santa basin is located. Furthermore, they highlight that GCMs have limited skill in simulating the interannual variability of rainy season onset and end, noting that many CMIP6 simulations still struggle to adequately represent the SAMS (see also Olmo et al., 2022). This suggests that the results from downscaled CMIP6 models and the downscaled CMIP5 models used in this study are consistent, at least based on the LM metric."**

Here are some areas of improvement:

**Comment #2.2**

**Line 14:** Introducing the bucket-type metric is a significant contribution but could be more prominently emphasized earlier in the abstract. Currently, it feels buried in the middle.

Thanks for this great suggestion. Implementing it (together with other comments) required some reorganization of the abstract. The full abstract now reads as:

**"In semi-arid regions, the timing and duration of the rainy season determines plant water availability, which directly impacts food security. Rainy season metrics, which aim to define and, in some cases, predict the onset and end of seasonal rains can support agricultural planning, such as scheduling planting dates and managing water resources. However, these metrics based on precipitation time series do not always accurately reflect plant water availability, and the variety of available metrics can complicate the selection of the most suitable one. Furthermore, a metric's ability to capture observed vegetation variability can indicate its applicability over larger spatial or temporal scales. This study introduces a new bucket-type metric that incorporates a simplified water balance, accounts for both accumulation and storage and also takes inter-annual legacy effects into account. We evaluate its performance against seven commonly used rainy season metrics, both calibrated and uncalibrated, using 18 years of satellite-derived Normalized Difference Vegetation Index from the semi-arid Rio Santa basin in the Peruvian Andes. Our results demonstrate that calibrating metrics using vegetation data significantly enhances their ability to capture rainy season dynamics, with the bucket metric outperforming others in both accuracy and robustness. Furthermore, we examine the sensitivity of all metrics to variations in rainfall intensity and frequency under future climate scenarios, using a previously published high-resolution dataset specifically designed for the Rio Santa basin which provides historical (1981–2018) rainfall data and future projections (2019–2100) based on 30 statistically downscaled CMIP5 models for RCP 4.5 and 8.5 scenarios respectively. While most rainy season metrics exhibit expected correlations in response to climatic changes, some established metrics display physically inconsistent behavior, likely due to methodological artifacts, highlighting their limitations in assessing hydroclimatic changes. In addition to the sensitivity analysis, we evaluate long-term trends in rainy season characteristics. Statistically downscaled CMIP5 ensemble projections for the future period suggest only a slight delay in the rainy season end, with no consistent trends in onset timing. Instead, inter-annual variability and ensemble spread remain the dominant influences. Our findings emphasize the need for careful calibration of metrics across diverse climate scenarios and different locations to ensure their reliability for agricultural planning, policymaking, and climate adaptation strategies. By providing a novel framework for evaluating rainfall metrics, this study offers a scalable approach that can be readily applied to other semi-arid regions."**

**Line 15:** The term "sensitivities" might need elaboration—does it refer to responsiveness, instability, or another issue?

We agree that this was unclear. We now changed this sentence, making our point clearer without being overly specific in the context of the abstract, l.15:

**"While most rainy season metrics exhibit expected correlations in response to climatic changes, some established metrics display physically inconsistent behavior, likely due to methodological artifacts, highlighting their limitations in assessing hydroclimatic changes."**

**Comment #2.4**

**Lines 40–45:** The discussion about the lack of strategies to validate rainy season metrics based on independent data is important but somewhat abrupt. Also, the emphasis on uncertainties in precipitation measurements is valid but could be expanded with specific examples of how these uncertainties affect the metrics or decision-making.

For example, tie the need for validation directly to the challenges of applying metrics in real-world scenarios. Add a sentence elaborating on the practical implications of these uncertainties, especially for agricultural or water management applications.

Thanks for noticing these issues. We added a sentence stating what practical consequences might follow due to uncertainties and the lack of a validation strategy, l. 41.

**"The resulting onsets and ends of rainy seasons can vary considerably depending on whether the methods were tailored to specific rain-gauge data, crop requirements or larger-scale characterization of temporal monsoon developments (Fitzpatrick et al., 2015; Sedlmeier et al., 2023).**

We also reorganized the following part of the paragraph to improve the reading flow, which now reads as l.43:

**"Often, the importance of determining rainy season characteristics for either agricultural planning, monitoring of ecosystems, assessments of temporal water availability in the light of a changing climate or water management topics in general is emphasized (e.g. Bombardi et al., 2019b; Fitzpatrick et al., 2015). However, precipitation time series are typically subject to significant uncertainties (e.g. Kidd et al., 2017; Pollock et al., 2018), which can lead water users and managers to make improper assumptions or take misguided actions. These uncertainties are particularly problematic in regions where decisions about planting, irrigation scheduling, or reservoir management rely heavily on short- or mid-term rainfall predictions. Furthermore, to the best of our knowledge, strategies to validate the outputs of rainy season metrics against independent observations are currently lacking. This raises concerns about whether such metrics**

**accurately capture conditions on the ground and highlights the need for validation frameworks that ensures their relevance and reliability for practical applications and allows to reliably deduce climatic changes. Furthermore, other aspects such as legacy effects beyond one hydrological year or the sensitivity of rainy season metrics to the alteration of the hydrological cycle, which is to be expected under climate change, have so far not been assessed."**

**Comment #2.5**

**Lines 57:** The transition to the Rio Santa Basin context is slightly abrupt, and the text does not adequately establish why this region is particularly suitable for testing the proposed framework.

We agree that the reading flow in this section was not ideal and that critical information, particularly regarding the semi-arid nature of the region, was lacking in our manuscript or was placed in the wrong place previously.

Inspired by your comment, along with your comment #2.9, that the climograph is not sufficiently utilized within the text, and a suggestion from the other reviewer (comment #1.3) that the review of existing methods could be more comprehensive, we decided to do some reorganization of the Introduction section. Specifically related to this comment, we moved the former Section 2.1 (Study Area) to the Introduction and provided a more thorough explanation of why the region is particularly relevant for testing our approach. By doing this, we now avoid repetition and create a more cohesive argument. Please refer to the revised Introduction as a whole. This particular change can be found from l.70:

**"In this study, we develop and demonstrate a novel approach to calibrate rainy season metrics using vegetation dynamics, focusing on the Upper Rio Santa basin (also: Callejón de Huaylas) in the tropical Peruvian Andes. This region is characterized by high seasonal variability of precipitation with the majority of annual precipitation occurring between September and April and little annual variability in temperature (see Figure 1 for the geographic location and a climograph). [...]"**

**Comment #2.6**

**Line 75:** The phrase "and two other precipitation datasets for comparison" is vague and could lead to confusion. It is unclear what "other" refers to—whether additional datasets are used for validation, alternative sources of precipitation data, or datasets of different spatial/temporal resolution.

Our aim here was just to briefly introduce what we are doing whereas the specific description of the data used is placed in the methods section below. We however agree that this might lead to confusion of readers. We now explicitly state which datasets we are referring to while still trying to stay brief, l.104:

**"[...] We combine several precipitation datasets with remote sensing data on temporal vegetation development. Specifically, we calculate the rainy season metrics based on convection-permitting, bias-corrected Weather Research and Forecasting (WRF) precipitation data and statistically downscaled CMIP5 projections (Potter et al., 2023) and use CHIRPS gridded data (Funk et al., 2015) as well as data from 3 local weather stations for comparison. [...]"**

**Comment #2.7**

**Lines 78:** While using NDVI for calibration is well-justified, it might help to briefly highlight why NDVI is a robust proxy for water availability in this region compared to alternatives.

We agree and revised this part, adding a statement to make it clearer why we consider NDVI to be a robust proxy, l.110:

**[...] Their research demonstrated that NDVI — an indicator of vegetation greenness available at high spatio-temporal resolution — captures variability and changes in water availability in the semi-arid Rio Santa basin, where water availability is the primary limiting factor for plant growth.**

**Comment #2.8**

**Lines 95:** The final objective includes exploring past and future changes in rainy season dynamics, but it does not specify the importance of these changes for the broader context of climate change. It should emphasize how understanding these changes can inform adaptation strategies in similar semi-arid regions.

We believe that the enumeration of the objectives itself should remain concise without too much discussing broader context. We however do agree that it is worthy to address the broader context and the topic of transferability more specifically. We therefore revised the paragraph right before the specific objectives and now emphasize more clearly the transferability and relevance in the context of adaptation strategies, l.117:

**"The principal objective of this study is to showcase a novel framework for characterizing the rainy season, emphasizing the importance of employing a calibration strategy for inferred rainy season onsets and ends. In addition, we test the sensitivity of rainy season metrics to plausible changes in rainfall intensity and frequency, as might occur due to global warming. By capturing shifts in seasonal rainfall dynamics, our approach provides a foundation for identifying and understanding hydrological changes that may inform future adaptation strategies. The proposed framework is designed to improve our understanding of variations in water availability within semi-arid regions, offering insights that extend beyond the Rio Santa basin and can be applied to similar climates. Regarding the Rio Santa basin, we aim to provide insights into past and future changes. We achieve this by: [...]"**

**Comment #2.9**

**Figure 1:** The climograph at the bottom is helpful but not explicitly referenced in the text. Discuss the precipitation seasonality and temperature trends shown in the climograph and connect them to the rainy season dynamics discussed in the study.

Recommendation: Add a sentence or two explicitly linking the Rio Santa basin's unique hydroclimatic and socioeconomic characteristics to the study's focus on rainy season metrics. Reference specific features of Figure 1 (e.g., NDVI, climograph) in the study area description to better integrate the figure with the text.

Indeed, we did not sufficiently reference the climograph in the text. Please refer to our answer regarding your Comment #2.5 for the specific implementation. Additionally, regarding the NDVI map, we now added a sentence to the revised introduction to underline the high spatial resolution which is a key argument for using it as a water availability proxy.

l.113: **"This high spatial resolution is shown in Fig. 1, which shows the 2000–2018 average NDVI for the Rio Santa basin, illustrating both longitudinal and altitudinal gradients."**

**Comment #2.10**

**Line 112 (Data):** The datasets have varying spatial resolutions. How do you account for or interpret these differences in your study?"

The main idea is that the high resolution NDVI data will be more reliable in the specific complex-topography setting of the Rio santa basin in comparison to coarser precipitation data. For this study specifically, we used spatial averages for all gridded datasets (and an average of 3 individual weather stations), thus eliminating the necessity of interpolation or regridding. That being said, during preprocessing we masked the gridded precipitation datasets (CHIRPS and WRF) to the areas where NDVI data is available, as shown in Fig. 1.

We already stated this for the LSP data, now slightly revised in l.142: *"**The resulting LSP data were averaged to the extent of the Rio Santa basin**."* and regarding the precipitation data we stated in l.158: *"Both gridded precipitation datasets were restricted to the geographical coverage of the available NDVI pixels as seen in Fig. 1 within the Rio Santa basin to acknowledge that high precipitation sums in the elevated Cordillera Blanca regions (e.g., glacierized or bare ground land-cover) do not align with vegetation responses and then spatially averaged (i.e. no spatial dimension)."* We believe this should now be clear for readers.

**Comment #2.11**

**Line 150-160:** The differentiation between onset-only metrics and metrics that address the onset and end of the rainy season is crucial but is introduced abruptly.

Recommendation: Include a transitional sentence to guide the reader through this distinction (e.g., "While some metrics are focused exclusively on the onset of the rainy season, others provide a more comprehensive approach by also addressing the season's end").

Thanks for noticing, we integrated your comment by combining your suggestion with information from our original sentence, l.183: **"While the FP and JD metrics are focused exclusively on the onset of the rainy season, the three remaining threshold-based metrics provide a more comprehensive approach by also addressing the end of the rainy season (hereafter RSE): [...]"**

**Comment #2.12**

**Line 179:** The formula is clear, but the conditional structure might be challenging for some readers to interpret.

Recommendation: Add a sentence explaining the formula in simpler terms: "The bucket water content is updated daily based on precipitation input and constant evapotranspiration. It is constrained between a minimum and maximum value, ensuring realistic water balance limits."

We added your suggestion to the section, l.206: **"Finally, we introduce a novel approach, which attempts to simulate a simplified water balance by consecutive balancing of daily input through rainfall and output through constant evapotranspiration, additionally constrained by a minimum and maximum bucket water content, ensuring realistic water balance limits:"**

**Comment #2.13**

**Section 3.3 Past and Future:** Transitioning from historical trends to future projections feels abrupt. Add a bridging sentence to guide readers, e.g., "Having established the variability in historical trends, we now turn to the projected changes in rainy season metrics up to 2100."

Your suggestion indeed increases the reading flow, thanks. We added a sentence to the transition: **"After establishing variability and trends for the historical period, we now explore the projected changes of rainy season metrics for the ensemble mean and standard deviation for each of the two RCP scenarios."**

**Comment #2.14**

Figures 6, 7, A7, and A8 are referenced, but their key findings are not fully summarized in the text.

Thank you, we agree that the text required some adjustments to better comprehend the key findings. To this end, we changed the first paragraph of the section which was directly addressing future projections but then immediately discussed past trends. We revised the first sentence to, l.362:

**"Finally, we calculated past metrics based on WRF data from 1981 to 2018 and projected future metrics up to 2100 using the statistically-downscaled CMIP5 model ensemble,**

**which comprises 30 individual models, and subsequently evaluated the trends for the historical and the future period."**

In the second paragraph, we believe there were too many details about methodological details limiting the visibility of the key results, such as the removal of certain models from further analysis. We effectively removed the information about how many models were discarded per metric and moved the part that described the strategy of removal to the methods part.

Besides, we reorganized and split up the paragraphs and added information. We believe the key results are now more visible, please refer to the section as a whole in the revised manuscript.

Regarding the supplementary figures A7 and A8, we tried to argue that only a minority of the CMIP5 models are actually responsible for the small trend we find in the case of the RSE of the bucket model. Indeed we do not show these numbers for all the insignificant trends but that is exactly why we introduced these supplementary figures for the more interested reader. We believe that listing the number of significant and non-significant individual models leading to an overall insignificant trend would not be interesting for potential readers. This statement as well as the supplementary figures mainly serve the purpose to point out that a reason why we cannot find convincing trends might be the lack of model agreement. To make this clearer, we now made small additions to the latter sentence of the second paragraph at l.384: **"An assessment of the distribution of significance of model trends for each metric and scenario can be found in Figs. A7 and A8."**

Despite these edits, we are unsure what else your comment is exactly referring to. If we overlooked something else, we kindly ask you to indicate the specific message we still may have failed to convey in the text.

**Comment #2.15**

**Line 361:** While uncertainties are acknowledged (e.g., in evapotranspiration rates), the potential impact on results is not fully explored. Provide more detail on how these uncertainties might affect the interpretation of trends.

In now l.415 we stated that the expected higher evapotranspiration rate will "*affect actual plant water availability and introduce uncertainty of currently unknown magnitude in the region*". We would like to stress that based on our analysis and current knowledge we can only speculate about the actual impacts. The combination of increased rainfall (as suggested by the CMIP5 models) with the increased evapotranspiration rates (through higher temperatures) are counteracting mechanisms. Potter et al., (2023) suggest an overall drying when taking evapotranspiration into account but they are only assessing potential evapotranspiration which is interesting to get a general idea about the direction of change but will deviate from actual evapotranspiration considerably. It is worth noting that, none of the rainy season metrics, being based on precipitation only, take changes in other variables affecting water availability into

account. Regarding this we specifically state in now l. 419 that *"the bucket metric is not intended to replace the tasks of sophisticated hydrological models, and realistically estimating actual evapotranspiration in a data-sparse environment is a complex task in itself."*

Therefore, one of the key messages for the future is that whenever such metrics are used in the context of water use (which many of them were developed for), things such as the increase in temperature and thus the evapotranspiration rate or other aspects such as higher frequency in droughts (or high intensity rainfalls) must be taken into account as well.

To better convey this idea, we added a sentence to the end of the paragraph (following the sentence we quoted above) putting more emphasis on this message, l. 421:

**"Meanwhile, it is therefore crucial to consider that when metrics like these are applied with water users in mind, factors beyond precipitation change (i.e. rising temperatures, wet/dry-spell frequency) must also be taken into consideration to ensure their practical relevance."**

**Comment #2.16**

The **conclusion** summarizes key findings well but could improve with clearer transitions, a detailed discussion of the bucket metric's strengths, and actionable practitioner recommendations.

1. While ENSO is mentioned, its significance to the study's findings could be elaborated further. Expand on how ENSO-related precipitation anomalies might influence the calibration and robustness of rainy season metrics.

   Thank you for your comment. Overall, we believe that precipitation anomalies, whether driven by ENSO or other factors, are unlikely to introduce significant issues with the calibration, provided that the rainfall distribution during the calibration period is representative. Regarding the representativeness of the calibration period (2000–2018) for the climate in the Rio Santa basin, we acknowledge the limited data availability for the region. However, the absence of historical trends in Figs. 6 and 7 suggests that it is reasonable to assume there is no significant long-term ENSO-related influence, as illustrated in the figure below. Furthermore, the figure demonstrates that precipitation anomalies respond non-linearly to the Niño 3.4 index, supporting the assumption that the effect of ENSO on the calibration is of lesser importance.

[Figure]

Figure for review 1: Seven-month running average of monthly precipitation sum anomalies for the Rio Santa basin domain using statistically-downscaled WRF data and unsmoothed 3-month shifted Nino 3.4 SSTa time series with El Niño/La Niña events being classified with a threshold value of ± 0.4.

One key point we aimed to highlight is that hydrological years following extreme anomalies present a limitation for all metrics, as they fail to account for legacy effects, thus giving reasoning for our new bucket approach. ENSO was used primarily as an illustrative example in this context. To clarify this, we have added the phrase **"for example"** in l.450, emphasizing that this issue is not specific to ENSO but applies to anomalies more broadly. However, we acknowledge that strong anomalies in the region are often associated with ENSO events.

Beyond these points, we believe that the response of individual metrics to anomalous hydrological years is already sufficiently explored in Section 3.2, even though it is not specifically tied to ENSO. Furthermore, in l.353 we already discuss the case of anomalous years which are way outside of the calibration range for the hydrological year 1989/1990 where:

*"[...] three metric outputs (Fig. 7a-c) are due to a dry spell lasting about three months leading to non-fulfilment of metric criteria and thus no-data labelling. Interestingly, LM and CB do not show any anomaly for this event because these metrics do not have information about any form of climatology. Conversely, this is accounted for by the bucket and threshold-based metric as the calibrated parameters represent the average climate of 2000 – 2018, such that extreme cases exceeding the calibration period cannot be informatively processed. We believe this is a desirable feature as for a practitioner this can be more informative than an unrealistic result in such cases."*

Interestingly, this event does neither co-incide with an El Nino or La Nina event, however it shows that the calibration is robust in the sense that in extreme cases no result will be produced which we believe is preferable over some odd result or as for the objective metrics which produce no anomaly as they have no information on the climatology.

Regarding the future projections, we want to emphasize that the uncertainty stems from the fact that the future overall development of ENSO is barely understood, making any projections inherently uncertain.

2. The flow between discussions of calibration, the bucket metric, and future projections could be smoother. Use transitional phrases to connect paragraphs, e.g., "Building on these findings, we introduced the bucket metric to address limitations in existing methods."

Motivated by your comment we improved the flow between the paragraphs. Specifically, we added these new beginnings to the corresponding paragraphs.

- **"Motivated by limitations in existing metrics, we designed our newly introduced bucket metric which outperforms…."** (l. 444)
- **"Using the bucket metric together with other calibrated and sensitivity-tested rainy season metrics and an unprecedented number of future projections, we conclude that…"** (l. 454)

3. The conclusions acknowledge uncertainty in projections but do not fully explore its implications for the study's findings. Discuss how these uncertainties might affect the interpretation of trends or the applicability of metrics in other regions.

Thank you for your comment. In this study, the uncertainties in projections are region-specific, as the dataset we use is confined to the Rio Santa basin. Consequently, we cannot address how uncertainties in projections might affect applicability in other regions. However, the metrics are designed with flexibility, making them broadly applicable in regions characterized by a unimodal distribution of seasonal rains. We already suggest this applicability several times, in the abstract (l.21), in the context of the overall validation framework (l.65), specifically for the bucket metric (l.446) and implicitly throughout the manuscript.

Regarding the interpretation of trends, the spread between models is substantial, with the majority indicating no significant changes in the onset or end of the rainy season. Depending on what specific metric is used, a considerable number of models suggest both positive and negative trends. This results in the ensemble predominantly showing no definitive trends while exhibiting a high standard deviation within the ensemble spread. We believe the interpretation in this context is straightforward: no clear trend emerges. While a trend might exist, it is likely obscured by the high level of uncertainty present in the projections.

Please also refer to our answer regarding your comment #2.14 on Figures A7/A8 where we address this in more detail and specifically discuss the implications of the ensemble spread and make corresponding revisions. Furthermore, your comment #2.15 also points in this direction and we made revisions according to that there as well. With this in mind, we hope you understand that we think this point is thoroughly addressed there and the conclusion would not benefit from repeating these arguments.

4. The reliance on vegetation proxies is discussed, but alternative calibration strategies are only briefly mentioned. Elaborate on potential alternative data sources, such as soil moisture or runoff measurements, and their advantages or challenges.

Thanks for your suggestion. We believe that this is an important point but in our opinion discussing aspects about alternative data sources we did not use in our analysis should not be part of the conclusions section. But as we agree that this is worth mentioning we added text to the introduction where we already made the point that a validation approach for rainy season metrics does not necessarily need to be based on vegetation indices, l.56:

**"This raises the challenge of designing an independent validation approach. While variables directly linked to the hydrological cycle such as soil moisture measurements would represent the ideal choice, their availability at (near-)climatological timescales is limited. In semi-arid regions, vegetation dynamics provide a useful alternative, as they exhibit a strong correlation with the seasonal precipitation cycle, albeit with a characteristic time lag (Hänchen et al., 2022). Remotely sensed proxies for vegetation development offer high spatiotemporal resolution and have been successfully used to study vegetation development for more than half a century (starting with Rouse Jr et al., 1973). We therefore argue that incorporating an independent metric validation scheme based on vegetation development provides three crucial advantages: [...]"**

5. The conclusion mentions practitioner needs but could provide more actionable recommendations. Include a sentence like, "Practitioners can use the bucket metric to better predict water availability in semi-arid regions, particularly during ENSO-driven precipitation anomalies."

Thanks for your suggestion, we added a new closing sentence in an more actionable sound but did not implement your suggestion regarding ENSO-driven precipitation anomalies as we believe our framework is suitable also in regions where seasonal precipitation is not driven by ENSO and this would be misleading, l.463:

**"Until then, practitioners as well as researchers can profit from more robust predictions of water availability building on our novel framework."**

6. The call for more robust climate models is appropriate but could be tied more explicitly to the study's findings. Link this need to specific challenges identified in the study, e.g., "The limitations of CMIP5 models, such as their inability to resolve ENSO impacts at finer scales, underscore the importance of developing high-resolution convection-permitting models."

Thank you for your suggestion. We want to point out that statistical downscaling generally does not allow to change the relationship between small-scale (such as locally observed) and large-scale (such as a climate model grid) weather and climate. As we

mentioned in our response to item 1. of this comment, the future development of ENSO is barely understood and moreover resolving its effects on finer scales is not the main issue we face. We revised our statement regarding the call for more robust climate models with your suggestions in mind by effectively splitting the two thoughts which were previously combined, I.459:

**"While our novel framework allows crucial insights derived from rainfall time series, an adequate assessment of future water availability for practitioners' needs would benefit from more robust climate model forcings, eventually to be expected from the emergence of high-resolution convection-permitting model projections, which will allow for a better representation of local precipitation. In addition, evapotranspiration changes should be further investigated, most appropriately analyzed through a sophisticated eco-hydrological model."**

---

## Referee Report (RR1)

The author present a nice study, evaluating existing metrics and a new metric for the onset and end of the rainy season with respect to vegetation data. One major result is that threshold based metrics needed to be calibrated region-specifically as they have done with the NDVI. Furthermore, they test sensitivities of the metrics, and they apply them to climate projections, showing that despite projected increase in precipitation, the rainy season onset and end do not show significant trends.

The study is interesting and relevant, however, the study could profit from an improved structure and some clarifications in the section Material & Methods as stated below in the specific comments..

Line 30: "hereafter named metrics" -> this seems a bit unnecessary. Maybe, it's fine if you just start the next sentence with: Broadly, these rainy season metrics…

Line 37: It's called the Standardized Precipitation Index (SPI), maybe denote it like this. Also, it is a bit a confusing citation. You are talking about RSO and RSE metrics, but the SPI is used to separate the rain season into wet and dry periods?  Consider removing this citation.

Line 46: it would be nice if you mention some examples of uncertainties here.

Line 61: How do you deal with irrigation when using a vegetation metric for independent validation?

Line 65: I think the sentence is nicer, if you remove "regarding such independent data"

Fig. 1: The topography colorscale here is not very intuitive. It could be useful to reduce the colorscale of topography to something simple, i.e. green to brown in order to give an intuitive idea of the topography

Line 145-162: I would suggest to first mention the three observational datasets (WRF, CHIRPS, AWS) and describe them shortly, and only then the scenarios. I.e. Move everything after "In addition, Potter et al. (2023)" to a later position in the paragraph.
It could be nice to mention the abbreviation that you use in the figures already here (WRF, CHIRPS, AWS).  For example, its' quite obvious that AWS means automatic weather stations, but it's never mentioned, I think. For reproducibility, could you state which three stations you used?

With respect to CMIP 5 downscaling, maybe it's good to mention briefly, what type of emission scenarios RCP4.5 and RCP 8,5 are.

Line 196: and the our -> "the" not needed?

Line 196-205: this small section is quite relevant. Maybe you could consider making an own small chapter out of it in a new chapter, for example: 2.2 Rainy season metrics, 2.3 The new "bucket" metric, 2.4. Calibration of threshold-based metrics, 2.5 Sensitivity analysis 2.6 Future projections. Also, in the Figures, you compare the thresholds provided by the authors to your calibrated

thresholds. Maybe, you can mention that you will be comparing them as well in the section on the calibration?

Are all the necessary information given, on how you apply the Differential Evolution optimization for reproducibility of your study?

Line 232: what is an almost large number of rainy seasons? Maybe consider deleting the "almost"?

Line 247: individual models were excluded individually -> maybe one individual is enough?

Sect. 2.4: It could also be helpful to separate the sensitivity analyses and projections in to separate chapter, because you will be doing two separate things with the climate projections? Do you state anywhere how you calculated the trends for the past and the future (maybe I just didn't see it)? That could go into such a short section?

Line 266: always add the unit after the RMSE, i.e. 8.8 and 14.4 days.

Fig 2 and 3: I find the figures nice, but very full. Could you consider either moving the calibration data or the RMSE (or both?) out of the Figure into a separate table?
Please mention more clearly that INIT WRF are the thresholds provided by the authors, maybe even mention this in the section very you describe the calibrations instead of just mentioning it in the caption?

Line 278ff: Here, mention that you talk about the INIT WRF  shown in  the figure, otherwiseit is not so clear. Also, specifically state the very high RMSE, when the parameters of the rainy season metrics are not tuned. I think this is a very relevant result of your study. For the agricultural perspective, a RMSE of 34 days (e..g Gurgiser) is very different from an RMSE of 12 days.
Also, could you state why you looked at the original threshold only using WRF, and not for example the AWS?

Fig 4: Please add inside the Figure a legend with the colors of the precipitation data sets. This can increase the readability a lot.

Lines 308 ff: You could also call Sect. 32. Sensitivity analysis of rain season metrics?

Fig. 5 This figure is quite small. Maybe you can enlarge the figures a bit by adding the rainy season metric title at the top of the figure (i.e horizontally). Like this you have more space in the horizontal dimension.

Conclusion: Could you add a sentence summarizing the results of the sensitivity tests?

Line 456: I don't think you can say it is an "unprecedented" number of future projections since these have been published by Potter already?